# Gilteritinib overcomes lorlatinib resistance in *ALK*-rearranged cancer

Hayato Mizuta[1,2,13], Koutaroh Okada[1,3,13], Mitsugu Araki [4], Jun Adachi[5], Ai Takemoto[1], Justyna Kutkowska [1], Kohei Maruyama[1,3], Noriko Yanagitani[6], Tomoko Oh-hara[1], Kana Watanabe[7], Keiichi Tamai[8], Luc Friboulet [9], Kazuhiro Katayama [10], Biao Ma [11], Yoko Sasakura[11], Yukari Sagae[4], Mutsuko Kukimoto-Niino[12], Mikako Shirouzu [12], Satoshi Takagi[1], Siro Simizu[2], Makoto Nishio[6], Yasushi Okuno [4], Naoya Fujita [1] & Ryohei Katayama [1,3 ✉]

*ALK* gene rearrangement was observed in 3%–5% of non-small cell lung cancer patients, and multiple ALK-tyrosine kinase inhibitors (TKIs) have been sequentially used. Multiple ALK-TKI resistance mutations have been identified from the patients, and several compound mutations, such as I1171N + F1174I or I1171N + L1198H are resistant to all the approved ALK-TKIs. In this study, we found that gilteritinib has an inhibitory effect on ALK-TKI–resistant single mutants and I1171N compound mutants in vitro and in vivo. Surprisingly, EML4-ALK I1171N + F1174I compound mutant-expressing tumors were not completely shrunk but regrew within a short period of time after alectinib or lorlatinib treatment. However, the relapsed tumor was markedly shrunk after switching to the gilteritinib in vivo model. In addition, gilteritinib was effective against *NTRK*-rearranged cancers including entrectinib-resistant NTRK1 G667C-mutant and *ROS1* fusion-positive cancer.

[1] Division of Experimental Chemotherapy, Cancer Chemotherapy Center, Japanese Foundation for Cancer Research, Tokyo, Japan. [2] Department of Applied Chemistry, Faculty of Science and Technology, Keio University, Kanagawa, Japan. [3] Department of Computational Biology and Medical Sciences, Graduate School of Frontier Sciences, The University of Tokyo, Tokyo, Japan. [4] Graduate School of Medicine, Kyoto University, Kyoto, Japan. [5] Laboratory of Proteomics for Drug Discovery, Laboratory of Clinical and Analytical Chemistry, Center for Drug Design Research, National Institutes of Biomedical Innovation, Health and Nutrition, Osaka, Japan. [6] Department of Thoracic Medical Oncology, The Cancer Institute Hospital, Japanese Foundation for Cancer Research, Tokyo, Japan. [7] Department of Respiratory Medicine, Miyagi Cancer Center, Miyagi, Japan. [8] Division of Cancer Stem Cell, Miyagi Cancer Center Research Institute, Miyagi, Japan. [9] INSERM U981, Gustave Roussy Cancer Campus, Université Paris Saclay, Villejuif, France. [10] Laboratory of Molecular Targeted Therapeutics, School of Pharmacy, Nihon University, Chiba, Japan. [11] Research and Development Group for In Silico Drug Discovery, Center for Cluster Development and Coordination (CCD), Foundation for Biomedical Research and Innovation at Kobe (FBRI), Kobe, Japan. [12] RIKEN Center for Biosystems Dynamics Research, Kanagawa, Japan. [13] These authors contributed equally: Hayato Mizuta, Koutaroh Okada. ✉email: ryohei.katayama@jfcr.or.jp

Abnormal fusion genes such as *ALK*, *ROS1*, and *NTRK* are commonly observed in subsets of patients with non-small cell lung cancer (NSCLC). Among these fusion genes, oncogenic *ALK* fusion genes are present in 3–5% of patients with NSCLC[1–5]. Although normal ALK protein activation is dependent on binding with its ligand[6], ALK fusion proteins oligomerize via the oligomerization domain of partner proteins, such as EML4, resulting in constitutive activation of ALK and its downstream pathways, thereby inducing tumorigenesis. To date, various *ALK* fusion genes have been reported, and the *EML4-ALK* fusion gene, which was first described in 2007 by Soda et al. is the most common form of *ALK* rearrangement[5,7,8].

The development of crizotinib, a first-generation ALK-tyrosine kinase inhibitor (TKI), revolutionized the treatment of ALK-positive NSCLC. Phase III clinical trials revealed that crizotinib is significantly superior to chemotherapy such as platinum agents combined with pemetrexed or docetaxel in terms of response rates and progression-free survival (PFS)[9,10]. However, cancer cells inevitably acquire drug resistance, resulting in tumor recurrence. Drug resistance mechanisms can be roughly categorized into ALK-independent and ALK-dependent processes. ALK-independent resistance mechanisms involve the activation of bypass pathways, such as EGFR, cMET, KRAS, and AXL or transformation into small cell lung cancer[11–17]. Contrarily, ALK-dependent drug resistance is associated with secondary mutations in ALK and/or the amplification of *ALK* fusion genes[13,18]. For example, the C1156Y, L1196M, and G1269A mutations alter the ATP-binding pocket structure and prevent crizotinib from binding to ALK[18,19]. To overcome crizotinib resistance, the second-generation ALK-TKIs alectinib, ceritinib, and brigatinib were developed, and they exhibited potent activity. Further, phase III clinical trials demonstrated that alectinib was associated with approximately 3-fold longer PFS than crizotinib in the first-line treatment of *ALK*-rearranged NSCLC. Thus, alectinib is widely used as a first-line therapy for *ALK*-rearranged NSCLC[20–23]. Unfortunately, as observed for crizotinib, almost all cancer cells acquired resistance to second-generation ALK-TKIs. Roughly half of the cases of resistance to second-generation ALK-TKIs involve secondary mutations in the ALK kinase domain[13,18]. In particular, G1202R and I1171N/S/T mutations frequently emerge after the failure of alectinib[18,24]. To overcome resistance associated with single mutations, the third-generation ALK-TKI lorlatinib was developed and approved[25–27]. In phase II clinical trials, lorlatinib produced an overall response rate of 47% and a median PFS of 7.3 months in the subset of ALK-positive patients who had received at least one ALK-TKI[26,28]. Further, a phase III study of lorlatinib versus crizotinib in the first-line setting is currently in progress[29]. However, several reports described compound mutations that induce lorlatinib resistance[30–36]. Interestingly, some compound mutations that lead to lorlatinib resistance result in re-sensitization to first- or second-generation ALK-TKIs. For example, C1156Y + L1198F and I1171N + L1256F led to re-sensitization to crizotinib and alectinib, respectively[31,34]. In addition, I1171N + L1198F mutants are more sensitive to crizotinib than I1171N single mutants[31]. Meanwhile, resistance to I1171N + L1196M can be overcome by ceritinib and brigatinib. Contrarily, other compound mutations including G1202R + L1196M, which is a solvent-front and gatekeeper mutation, result in high resistance to all ALK-TKIs[36]. Thus, the identification of novel agents to overcome these various resistance mutations including highly drug-resistant compound mutations is strongly needed.

In this study, we aimed to identify agents to address all known mutations conferring resistance to clinically approved ALK-TKIs, namely I1171N + F1174I and I1171N + L1198H. Via inhibitor library screening, we found that gilteritinib, a TKI approved for treating relapsed or refractory FLT3-positive acute myeloid leukemia (AML), can overcome these resistance mutations. Then, we investigated whether gilteritinib has inhibitory effects on other single or compound mutations. Using cell viability assays and western blot analysis, the efficacy of gilteritinib against cells carrying various single and compound mutations, especially I1171N, was demonstrated. These results were confirmed using patient-derived cells in vitro and in vivo. In addition, we investigated the efficacy against TKI-resistant mechanisms mediated by bypass pathway activation such as KRAS G12C mutation and AXL activation. Furthermore, the inhibitory activity of gilteritinib against *ROS1*-rearranged cancer was equivalent to that of crizotinib, the approved TKI for *ROS1*-rearranged NSCLC. Moreover, gilteritinib effectively inhibited the NTRK1 fusion protein and suppressed the growth of *NTRK*-rearranged cancer. Gilteritinib additionally displayed efficacy against cells carrying the NTRK G667C mutation, which reportedly confers resistance to the approved NTRK inhibitor entrectinib.

## Results

**Identification of gilteritinib as a drug that overcomes lorlatinib-resistant compound mutants**. In our previous report, we discovered various lorlatinib-resistant EML4-ALK compound mutants and found that some of these mutants were re-sensitized to clinically approved ALK-TKIs using in vitro and in vivo experiments. However, EML4-ALK I1171N + F1174I and I1171N + L1198H compound mutants were resistant to all approved ALK-TKIs.

To identify drugs that can overcome these mutations, we screened our focused 90-inhibitor library using Ba/F3 cells expressing EML4-ALK I1171N + F1174I and I1171N + L1198H. At a concentration of 50 nM, gilteritinib suppressed the viability of both wild-type (WT) and compound mutant Ba/F3 cells (Fig. 1a). Since gilteritinib has reported as multi-kinase inhibitor, we first evaluated whether the ALK inhibitory efficacy of gilteritinib is driven by on-target activity or not. As a result of western blot analysis, gilteritinib suppressed the autophosphorylation of ALK in EML4-ALK-expressing Ba/F3 cells (Fig. 1b and Supplementary Fig. 1). Further, compared with approved ALK-TKIs, gilteritinib showed much lower $IC_{50}$ to EML4-ALK I1171N + F1174I and I1171N + L1198H mutant-expressing Ba/F3 cells (Supplementary Table 1). As well as Ba/F3 cell model, we verified the activity of gilteritinib to H3122 cell line that is EML4-ALK-positive NSCLC cells. While alectinib, lorlatinib, and gilteritinib suppressed the ALK phosphorylation and the downstream pathway in H3122 parental cells, only gilteritinib showed the inhibitory effect to I1171N + F1174I mutant overexpressed H3122 cells at low concentration (Fig. 1c and Supplementary Fig. 2). Then, the inhibitory concentration of gilteritinib is matched between ALK phosphorylation and its downstream signals. Furthermore, to confirm whether gilteritinib directly inhibit ALK, we performed in vitro kinase assay. The ALK kinase activity was inhibited in dose-dependent manner, and $IC_{50}$ was shifted by the increasing concentration of ATP. These results suggested gilteritinib competitively inhibited ALK tyrosine kinase (Fig. 1d). Additionally, phosphoproteomic analysis of ALK-positive lung cancer cells with or without gilteritinib treatment revealed that gilteritinib significantly decreased phosphorylation of ALK and its adapter proteins such as IRS1/2, SOS2, or SH2B1. Further, using the phosphoproteomics data, kinase substrate-enrichment analysis (KSEA) predicted activate kinase candidates as potential gilteritinib targets and identified ALK as the most likely regulated protein (Fig. 1e, Supplementary Figs. 3 and 4, and Supplementary Table 2). These results indicated that gilteritinib directly inhibit ALK in ATP competitive manner and inhibit the growth of *ALK*-rearranged cancer cells.

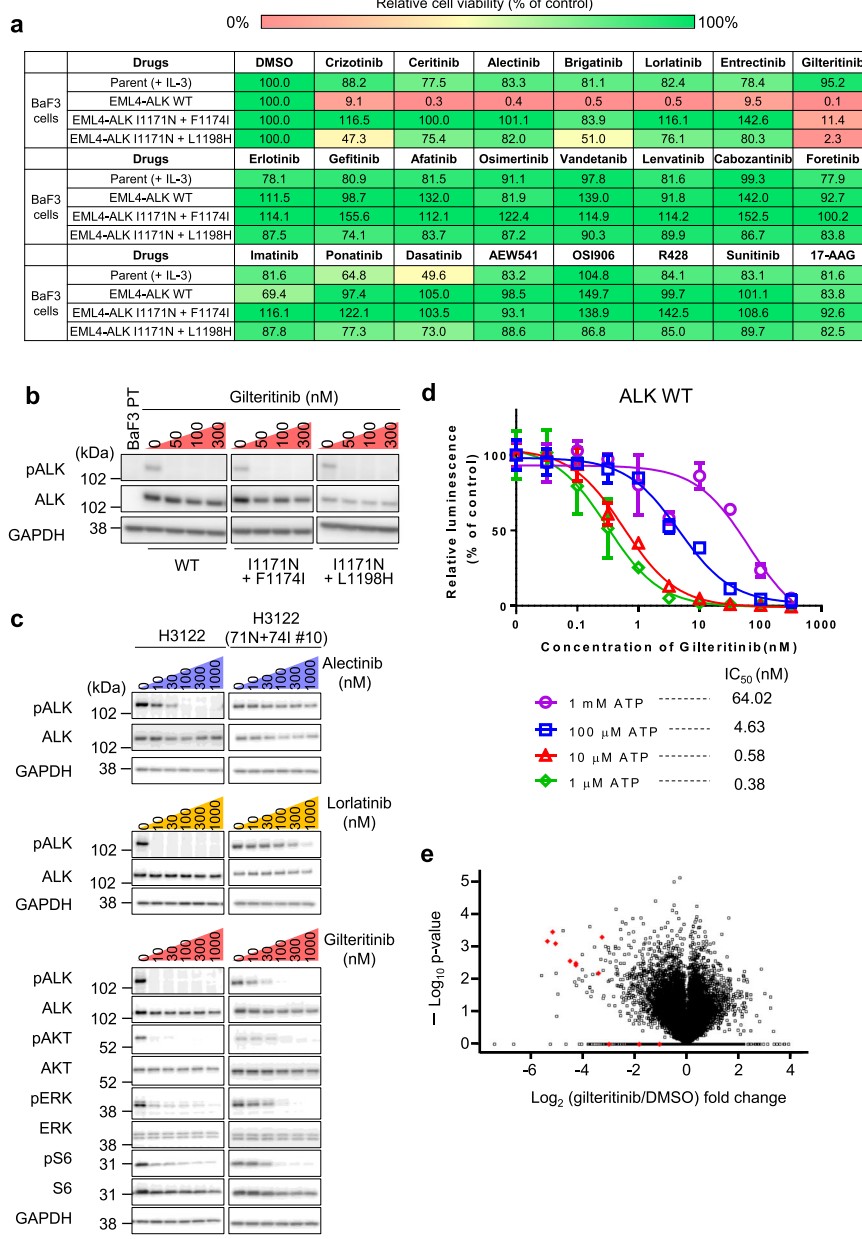

**Fig. 1 Identification of gilteritinib as an inhibitor of *ALK*-rearranged cancer cells. a** Relative cell viability of parental (with IL-3), EML4-ALK wild-type (WT), EML4-ALK I1171N + F1174I, and EML4-ALK I1171N + L1198H Ba/F3 cells treated with 50 nM of the indicated inhibitors. Cell viability was analyzed using the CellTiter-Glo assay and calculated relative to the viability of dimethyl sulfoxide-treated Ba/F3 cells. **b** The suppression of phospho-ALK in I1171N + F1174I, I1171N + L1198H mutation-expressing Ba/F3 cells was evaluated using western blotting. Cells were treated with the indicated concentrations of gilteritinib for 3 h (*n* = 2). **c** The suppression of phospho-ALK and its downstream signals in H3122 parental cells and I1171N + F1174I compound mutation-expressing cells were evaluated using western blotting. Cells were treated with the indicated concentrations of inhibitors for 6 h (*n* = 2). **d** The evaluation of the inhibitory activity of gilteritinib in the in vitro kinase assay using the ADP-Glo assay kit showed a dose-dependent decrease in ALK activity with gilteritinib according to the increase of ATP concentration. *N* = 3 independent samples examined over three independent experiments and representative experiment data are presented as mean values ± SD. **e** Volcano plot displaying the −log₁₀ (*p*-value) versus log₂ (gilteritinib treatment/DMSO treatment) for all quantified phosphopeptides. The red diamond indicates phospho-ALK.

**Gilteritinib exhibited potent antitumor activity against patient-derived cancer cells**. Next, we checked whether gilteritinib displayed antitumor activity against *ALK*-rearranged human NSCLC cells. Using cell viability assays, we confirmed the potent activity against *EML4-ALK* fusion gene (WT, v1, and v3)-positive NSCLC cells (H3122 and H2228 cells) and patient-derived primary cancer cells (JFCR-028-3 and JFCR-018-1) (Fig. 2a and Supplementary Table 3A). Furthermore, gilteritinib exhibited efficacy against *NPM-ALK* fusion oncogene-positive KARPAS 299 cells, a non-Hodgkin's Ki-positive large cell lymphoma cell line (Fig. 2a and Supplementary Table 3A). ALK autophosphorylation in these *ALK*-rearranged human cancer cell lines was also suppressed by gilteritinib (Fig. 2b and Supplementary Fig. 5). Consistently with the above data, flow cytometry analysis illustrated that gilteritinib induced marked apoptosis in H3122 cells (Fig. 2c). However, the drug had minimal or no effects in PC9, HCC827 (harboring EGFR

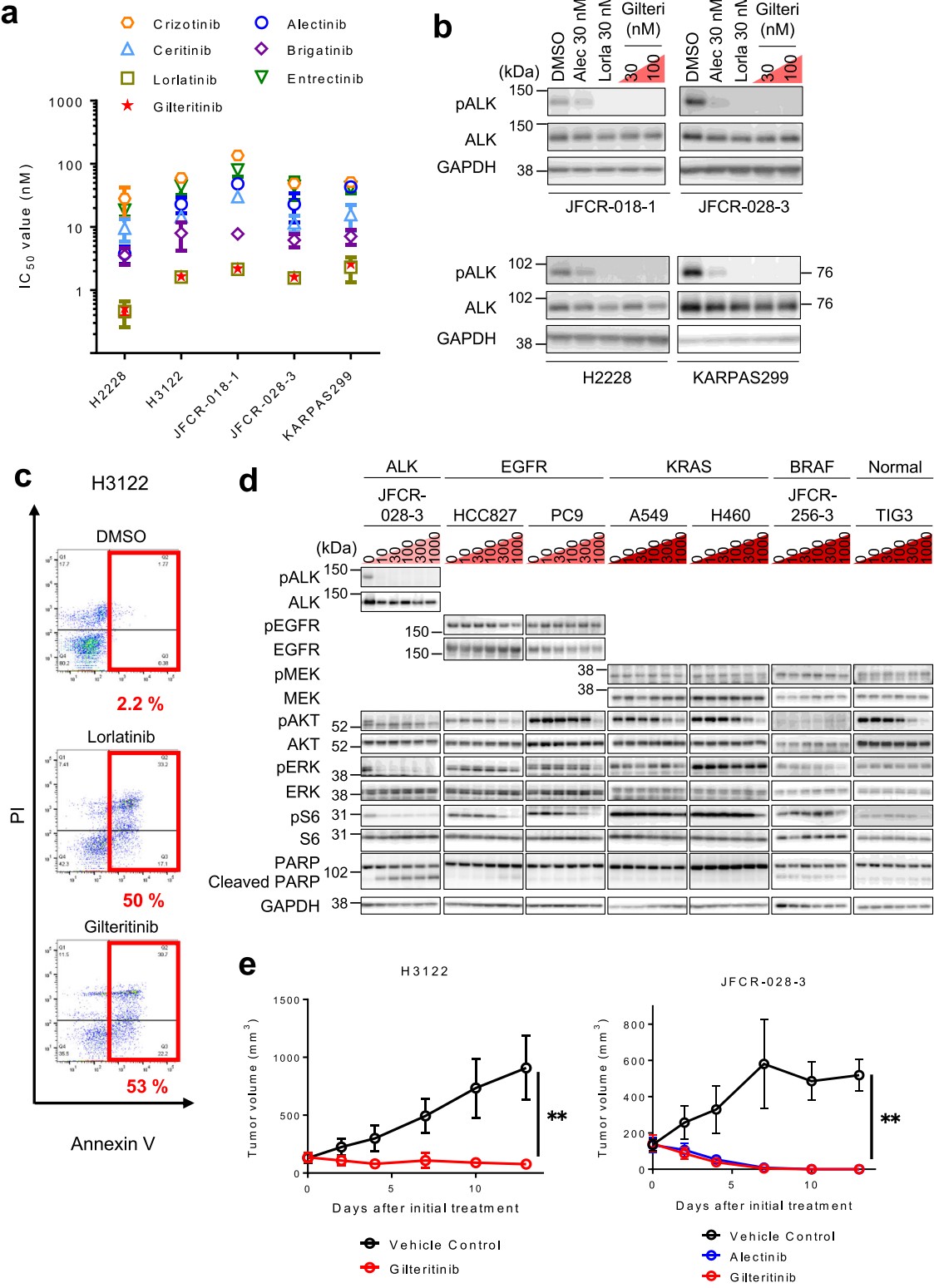

active mutant), H460, A549 (KRAS mutation-positive), JFCR-256-3 (BRAF mutation-positive patient-derived cancer cells), and TIG-3 cells (normal human lung fibroblasts) (Fig. 2d and Supplementary Figs. 6, 7).

As our in vitro studies demonstrated the potent antitumor activity of gilteritinib to *ALK*-rearranged cancer cells, we subsequently performed an in vivo study. As expected, treatment with

30 mg/kg gilteritinib-induced tumor shrinkage in mice carrying WT EML4-ALK harboring JFCR-028-3 or H3122 tumors without body weight loss (Fig. 2e and Supplementary Fig. 8A, B).

**Antitumor activity of gilteritinib against ALK-resistant mutants to first-generation or second-generation ALK-TKIs.** Previous reports revealed that secondary mutations in the ALK

**Fig. 2 Efficacy of gilteritinib in patient-derived NSCLC cells. a** $IC_{50}$ calculated from the viability analysis of *ALK* fusion gene-positive cancer cell lines and patient-derived primary cancer cell lines. Cells were treated with the indicated inhibitors for 72 h. $N = 3$ independent samples examined over three independent experiments and data presented as mean values ± SD. **b** The suppression of phospho-ALK in *ALK* fusion gene-positive cancer cell lines and patient-derived primary cancer cell lines was evaluated via western blotting. Cells were treated with the indicated concentrations of gilteritinib for 6 h ($n = 2$). **c** Flow cytometric analysis of apoptosis using Annexin-V and propidium iodide staining after 72 h of treatment with 100 nM lorlatinib or gilteritinib in H3122 cells. The percentage of cells undergoing apoptosis is shown in red. **d** The suppression of phospho-RTKs and its downstream signals in indicated NSCLC cells and TIG3 cells were evaluated using western blotting. Cells were treated with the indicated concentrations of gilteritinib for 6 h ($n = 2$). **e** H3122 (left) and JFCR-028-3 cells (right) were subcutaneously transplanted into BALB/c *nu/nu* mice. When the average tumor volume reached ~150 mm³, the mice were randomized to treatment with vehicle control, alectinib (30 mg/kg), or gilteritinib (30 mg/kg) treatment group once daily for 5 days/week via oral gavage ($n = 6$ per treatment group). Tumor volumes were measured three times a week. Data are presented as mean values ± SD. The significance of differences on day 13 was calculated using the two-sided Mann–Whitney $U$ test. (*P* value: Vehicle vs. gilteritinib treatment group (H3122), 0.0022; Vehicle vs. gilteritinib-treatment group (JFCR-028-3), 0.0048).

kinase domain such as I1171T/N/S, V1180L, G1202R, and L1196M emerged in patients with alectinib-resistant cancer. The $IC_{50}$ of gilteritinib in cells carrying these mutants, excluding G1202R, was <30 nM (EML4-ALK I1171T, 4.17 nM; EML4-ALK I1171N, 6.13 nM; EML4-ALK I1171S, 2.86 nM; EML4-ALK V1180L, 1.45 nM; EML4-ALK L1196M, 20.4 nM; EML4-ALK G1202R, 168 nM). In addition to these alectinib-resistant mutants, the growth of other mutants known to be resistant to crizotinib or ceritinib was also inhibited by gilteritinib excluding D1203N ($IC_{50}$: EML4-ALK C1156Y, 0.66 nM; EML4-ALK F1174V, 3.41 nM; EML4-ALK F1245V, 1.41 nM; EML4-ALK G1269A, 1.39 nM; EML4-ALK T1151K, 1.24 nM; EML4-ALK F1174I, 4.72 nM; EML4-ALK L1196Q, 25.5 nM; EML4-ALK D1203N, 53.0 nM). In our recent work, we discovered the novel lorlatinib-resistant mutant EML4-ALK L1256F. The growth of this single mutant was also inhibited by gilteritinib ($IC_{50} = 0.34$ nM; Fig. 3a and Supplementary Table 1). Of note, ALK autophosphorylation was completely attenuated by 50 nM gilteritinib treatment except for solvent-front mutation, G1202R and D1203N (Fig. 3b).

In addition to Ba/F3 cells expressing ALK-TKIs-resistant mutations, we confirmed that gilteritinib inhibited the cell growth and induced apoptosis of alectinib-resistant patient-derived MCC-003 cells harboring the EML4-ALK-I1171N mutation at similar concentrations as ceritinib, brigatinib, and lorlatinib (Fig. 3c, d and Supplementary Table 3A). Next, we performed an in vivo study using MCC-003 cells. Whereas marked tumor regression was induced in mice treated with gilteritinib, tumor regression was not identified in alectinib-treated animals (Fig. 3e). We did not observe significant body weight loss in either group (Supplementary Fig. 8C). To check whether gilteritinib down-regulates phospho-ALK in vivo, we performed immunoblot analysis to detect the autophosphorylation of ALK kinase and its downstream signaling molecules in MCC-003 tumors. The results illustrated that gilteritinib inhibited phospho-ALK and its downstream signaling (Fig. 3f).

**Efficacy of gilteritinib against lorlatinib-resistant EML4-ALK compound mutants identified in the clinic.** Lorlatinib has the potent inhibitory effect against almost all first-generation and second-generation ALK-TKIs-resistant single mutants. However, previous research demonstrated that compound mutations in ALK kinase domain were emerged and induced resistance to lorlatinib[34,36]. We recently experienced a patient carrying the ALK I1171S + G1269A compound mutation (Fig. 4a). This patient received chemotherapy (cisplatin/pemetrexed/bevacizumab for four cycles) followed by ALK-TKIs including crizotinib, alectinib, and lorlatinib. A resistance mutation was identified in metastatic liver cancer after lorlatinib treatment. In our previous paper, we reported that the ALK I1171S + G1269A compound mutant was sensitive to the second-generation

ALK-TKIs ceritinib and brigatinib, and tumor regression was achieved in this patient after ceritinib treatment.

In this research, we evaluated whether gilteritinib has potential activity against various lorlatinib-resistant compound mutants. Our cell viability assay and western blotting demonstrated that gilteritinib was effective against ALK I1171S + G1269A compound mutant, as well as I1171S single mutant ($IC_{50}$: I1171S + G1269A, 13.3 nM; I1171S, 2.86 nM; Fig. 4b–d and Supplementary Table 1). Furthermore, gilteritinib had a relatively low $IC_{50}$ (<30 nM) in all tested I1171N compound mutant Ba/F3 cells (I1171N + F1174I, 23.5 nM; I1171N + F1174L, 3.15 nM; I1171N + L1196M, 14.0 nM; I1171N + L1198F, 1.64 nM; I1171N + L1198H, 6.95 nM; I1171N + L1256F, 0.41 nM; I1171N + G1269A, 11.4 nM; Fig. 4c, d and Supplementary Table 1). To further explore the activity of gilteritinib against the lorlatinib-resistant compound mutation, we established ALK I1171N + F1174I compound mutant overexpressed JFCR-028-3 cells and treated with alectinib, lorlatinib, or gilteritinib in vivo. While the tumor regrowth within short period was observed in both alectinib and lorlatinib-treated mice, gilteritinib-treated mice showed complete remission of tumor over 50 days. In addition, after the tumor regrowth on lorlatinib or alectinib treatment, gilteritinib were administered to those alectinib or lorlatinib-resistant tumor-bearing mice. Surprisingly, the immediate tumor shrinkage was observed by switching to gilteritinib treatment (Fig. 4e and Supplementary Fig. 9). These results strongly indicated the potent efficacy of gilteritinib against all clinically approved ALK-TKIs-resistant compound mutants.

On the other hands, we also experienced ALK G1202R + L1196M and D1203N + F1245V compound mutants harboring patients whose tumors relapsed after sequential ALK-TKIs therapy including lorlatinib. A schematic of the timing of treatment and biopsy is presented in Fig. 5a. In addition, a recent report by Recondo et al. identified several ALK compound mutations including L1196M + D1203N that conferred high resistance to all clinically available ALK-TKIs[33]. Thus, we tested the sensitivity of these mutants to gilteritinib. Different from I1171N/S compound mutants, gilteritinib was less effective against the G1202R + L1196M, D1203N + F1245V, and D1203N + L1196M compound mutants ($IC_{50}$: G1202R + L1196M, 117 nM; D1203N + F1245V, 64 nM; L1196M + D1203N, 109 nM; Fig. 5b–d and Supplementary Table 1). Gilteritinib also failed to effectively suppress ALK autophosphorylation in these mutants (Fig. 5e and Supplementary Fig. 10).

**Prediction of binding mode of gilteritinib to ALK by the computational simulation.** To predict the binding mode of gilteritinib to ALK, we performed computational simulations based on the crystallographic information of Fms-like tyrosine kinase 3 (FLT3) complexed with gilteritinib (PDBID: 6JQR), assuming

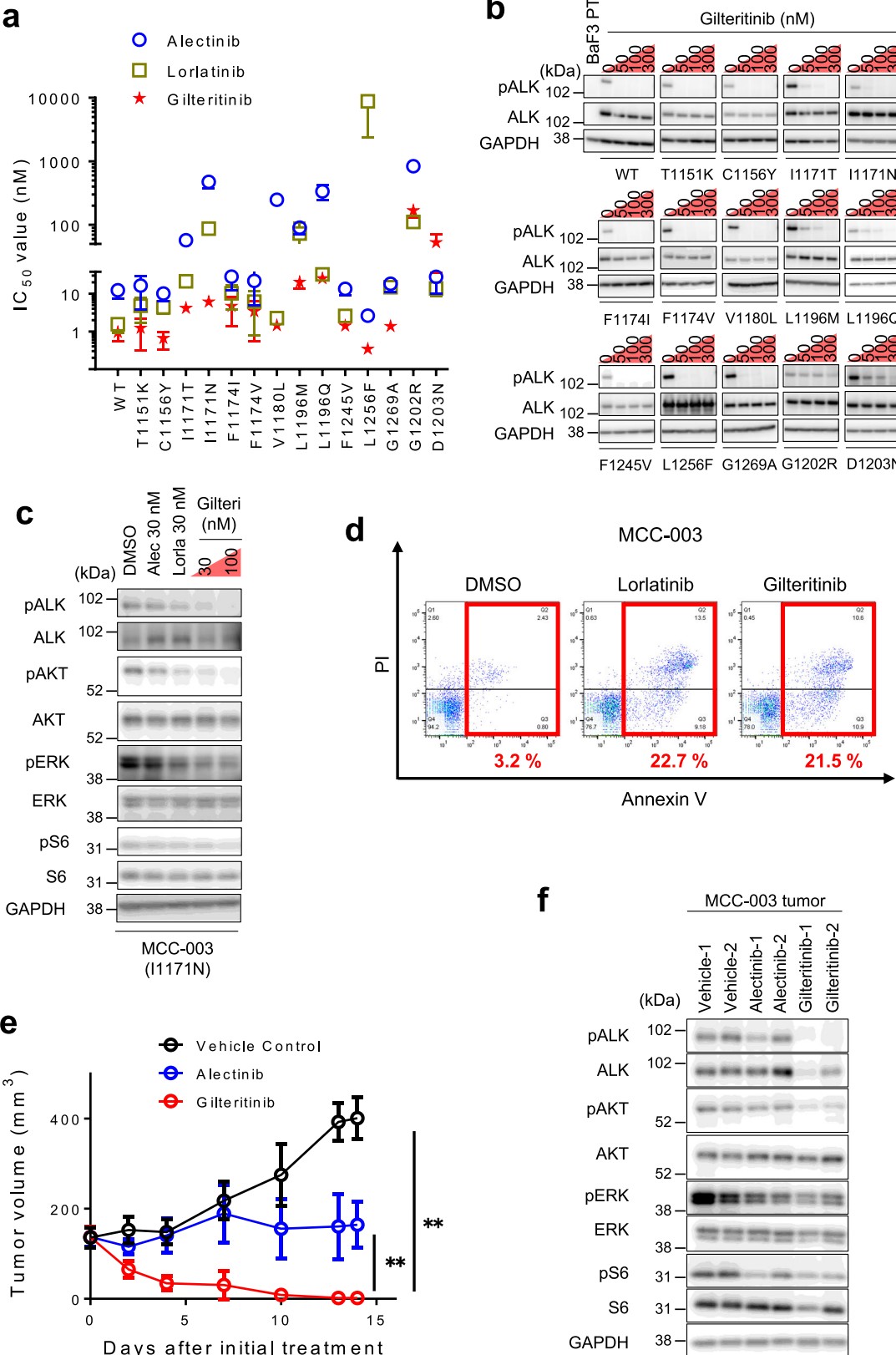

that gilteritinib has a similar binding geometry between ALK and FLT3. Our molecular docking and molecular dynamics (MD) simulation indicated that gilteritinib fitted into the ATP-binding pocket of ALK (Fig. 6a and Supplementary Fig. 11), and three hydrogen bonds between E1197, M1199, and E1210 of ALK and

gilteritinib were formed (Fig. 6b). In our recent paper, we performed free energy simulations using MP-CAFEE and successfully quantified the sensitivities of ALK inhibitors to multiple ALK-TKI-resistant mutants, showing a clear linear correlation between the experimental $IC_{50}$ and the binding free energy ($\Delta G$).

**Fig. 3 Efficacy of gilteritinib against first- or second-generation ALK-TKI–resistant single mutants. a** IC$_{50}$ calculated from the viability analysis of Ba/F3 cells carrying single mutations conferring resistance to first-generation or second-generation ALK-TKIs. Cells were treated with lorlatinib, alectinib, or gilteritinib for 72 h. $N = 3$ independent samples examined over three independent experiments and data presented as mean values ± SD. **b** The suppression of phospho-ALK in Ba/F3 cells carrying single mutations conferring resistance to first-generation or second-generation ALK-TKIs was evaluated via western blotting. Cells were treated with the indicated concentrations of gilteritinib for 3 h ($n = 2$). **c** The suppression of phospho-ALK and its downstream signals in MCC-003 cells harboring EML4-ALK I1171N was evaluated via western blotting. Cells were treated with the indicated concentrations of drugs for 6 h ($n = 2$). **d** Flow cytometric analysis of apoptosis using Annexin-V and propidium iodide staining after 72 h of treatment with 100 nM lorlatinib or gilteritinib in MCC-003 cells. The percentage of cells undergoing apoptosis is shown in red. **e** MCC-003 cells were subcutaneously transplanted into BALB/c *nu/nu* mice. When the average tumor volume reached ~150 mm$^3$, the mice were randomized to treatment with vehicle control, alectinib (30 mg/kg), or gilteritinib (30 mg/kg) treatment group once daily for 5 days/week via oral gavage. Tumor volumes were measured three times a week ($n = 8$ per treatment group). Data are presented as mean values ± SD. The significance of differences on day 14 was calculated using the two-sided Mann–Whitney $U$ test ($P$ value: Vehicle vs. gilteritinib treatment group, 0.0002; Alectinib vs. gilteritinib treatment group, 0.0002). **f** Phospho-ALK and its downstream signals in MCC-003 tumor samples obtained from vehicle-, alectinib-, or gilteritinib-treated mice were evaluated via western blotting ($n = 2$).

Since our free energy simulations could correctly predict how each ALK-TKI-resistant mutation affects the drug-binding[31], we challenged to apply this simulation method to estimate the binding affinity of gilteritinib with the ALK-TKI-resistant mutants. As the results, the calculated ΔG was correlated well with the experimental IC$_{50}$ obtained from Ba/F3-EML4-ALK-mutant cells ($R = 0.627$; Fig. 6c). As above described, gilteritinib has less potency against ALK-G1202R mutant, and our computational simulations suggested that the mutated arginine residue clashes with the methoxy group of gilteritinib, resulting in a significant decrease in the binding affinity due to loss of inter-molecular electrostatic interactions (Supplementary Fig. 12A). Conversely, gilteritinib exhibited increased affinity for L1198F-mutant EML4-ALK, as indicated by the difference in IC$_{50}$ versus the WT (0.1 vs. 0.78 nM; Fig. 6d and Supplementary Table 1). Consistently, Ba/F3 cells carrying the L1198F + G1202R compound mutation were more sensitive to gilteritinib than G1202R-mutant cells (IC$_{50}$: L1198F + G1202R, 32 nM; G1202R, 168 nM; Fig. 6d and Supplementary Table 1), and cells carrying the I1171N + L1198F mutation were more sensitive to gilteritinib than I1171N-mutant cells (IC$_{50}$: I1171N + L1198F, 1.6 nM; I1171N, 6.1 nM; Supplementary Table 1). Using in vitro kinase assay, we further confirmed that ALK L1198F was more sensitive to gilteritinib similar to the crizotinib (Fig. 6e). Also, our computational simulations suggested that L1198F mutation enhanced the binding affinity to gilteritinib by increasing both van-der-Waals and electrostatic interactions (Supplementary Figs. 12B, C and 13).

**Overcoming the bypass pathways activation-mediated ALK-TKI resistance by gilteritinib.** As previously reported, ALK-TKI-resistant mechanisms are roughly categorized into ALK-dependent or ALK-independent, so-called bypass pathway activation such as EGFR and MET. Recently, Taniguchi et al. demonstrated activated AXL is associated with a low response to EGFR-TKIs and the emergence of drug-tolerant cells in EGFR-positive lung cancer[37]. Similarly, previous reports suggested AXL is one of the key molecular to acquire resistance to ALK-TKIs. As gilteritinib can inhibit AXL, we evaluated whether gilteritinib prevent development of ALK-TKI resistance via AXL activation. At a concentration of 50 nM, both alectinib and gilteritinib suppressed ALK phosphorylation in AXL overexpressed H3122 cells. However, compared with gilteritinib treatment, AXL phosphorylation and downstream signaling pathways such as MAPK pathway and PI3K-AKT pathway were still activated in alectinib treatment (Supplementary Fig. 14A). Further, AXL expression induced noticeable increase of IC$_{50}$ of alectinib but that of gilteritinib remained <5 nM (Supplementary Fig. 14B). Next, we evaluated the efficacy of gilteritinib by mouse xenograft tumor model using AXL-overexpressed H3122 cells. Alectinib treatment

showed partial tumor growth suppression, but about after 3 weeks, tumor regrowth was observed. As alectinib-treated H3122 parental tumors did not regrow during the same period, it was suggested that observed resistance was dependent on the overexpressed AXL. On the other hand, gilteritinib significantly suppressed tumor growth and the size of tumors was maintained for more than 5 weeks (Supplementary Fig. 15A–C). Moreover, to evaluate the efficacy of gilteritinib against the tumors that showed resistance to alectinib, the alectinib-treated mice were randomized into two groups and gilteritinib was administered to one group, and the other group was continuously administered alectinib. As a result, gilteritinib treatment clearly inhibited the tumor growth compared with the continuous alectinib treatment (Supplementary Fig. 15A).

Next, we evaluated the KRAS signaling pathway. Previously, it was reported that the lineage switching and activation of the MAPK pathway is one of the important mechanisms of resistance to ALK-TKIs[16,17]. Further, using single-cell RNA sequencing, Maynard et al. recently revealed the EML4-ALK-positive tumor sample from the patient after multiple lines of ALK-TKI and other therapy contained *KRAS* G12C and *KRAS* G13D mutations[38]. Thus, in this study, we focused on whether *KRAS* G12C mutation serve as resistant mechanism against gilteritinib. To imitate this situation, KRAS G12C was introduced into two patients derived ALK-positive cells, JFCR-028-3 and MCC-003. The IC$_{50}$ of KRAS G12C-expressing MCC-003 was significantly increased and western blot analysis demonstrated gilteritinib treatment could not completely suppress the downstream signaling pathways (Supplementary Fig. 16A, B). Of note, KRAS G12C overexpressed JFCR-028-3 cells, gilteritinib partially suppressed the downstream signaling molecules (Supplementary Fig. 16C). To overcome the KRAS G12C-mediated resistant, we evaluated the combination therapy of gilteritinib and KRAS G12C-specific inhibitor, AMG510 in vitro and in vivo. As a result, combined treatment with gilteritinib and AMG510 inhibited downstream pathways and showed low IC$_{50}$ (Supplementary Fig. 16A–D). In vivo analysis consistently demonstrated whereas both single agents could not completely suppress tumor growth, combined treatment significantly induced tumor regression (Supplementary Fig. 17).

Finally, we checked EGFR signaling pathway. As well as alectinib, gilteritinib failed to inhibit cell growth of JFCR-098 cells that was obtained from the patient observed ALK-TKI-resistant via EGFR pathway. However, JFCR-098 cells showed high sensitivity to combined treatment with gilteritinib and EGFR-TKI afatinib (Supplementary Fig. 18).

Overall, our results indicated gilteritinib is effective to prevent acquiring resistance via AXL signaling by single agent, and combination strategy targeting activated bypass pathways such as EGFR or KRAS is also effective.

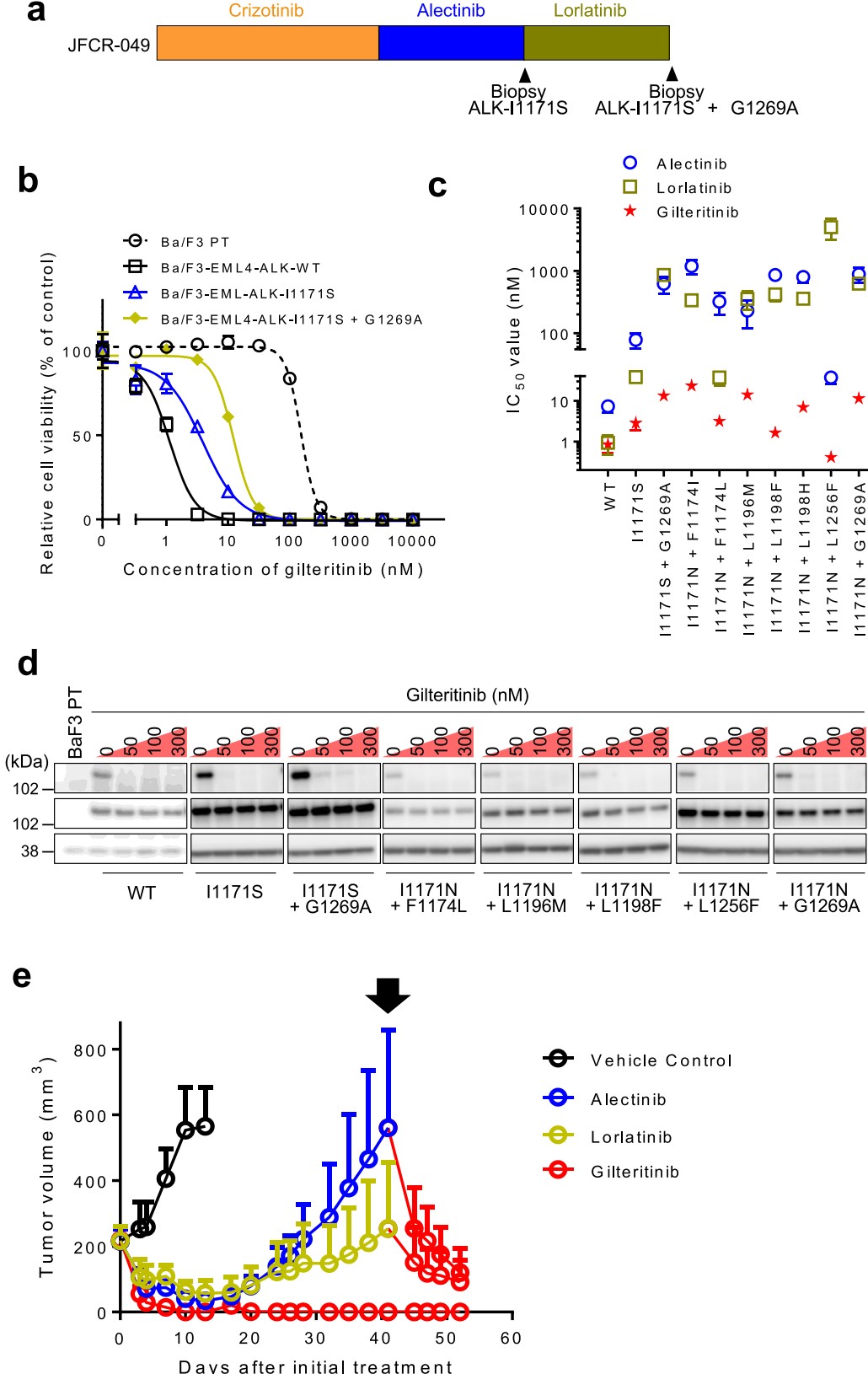

**Effectiveness of gilteritinib against *ROS1*-rearranged and *NTRK1*-rearranged cancers.** Many oncogenic driver genes have been identified in NSCLCs. *ROS1* or *NTRK* rearrangement accounts for ~1% and 0.1% of NSCLCs, respectively. Two first-generation

NTRK-TKIs (entrectinib and larotrectinib) are clinically approved for the treatment of *NTRK*-rearranged cancers. Because the tyrosine kinase domains of ROS1 and NTRK share structural similarity with that of ALK and multiple ALK inhibitors exert inhibitory effects on

**Fig. 4 Efficacy of gilteritinib against EML4-ALK I1171N compound mutants. a** Schematic of the timing of ALK-TKI treatment and biopsy for patients from whom the JFCR-049. **b** The inhibitory activity of gilteritinib in indicated EML4-ALK expressing Ba/F3 cells. Cells were treated with inhibitors for 72 h and analyzed cell viability using the CellTiter-Glo assay. $N = 3$ independent samples examined over three independent experiments and representative experiment data are presented as mean values ± SD. **c** $IC_{50}$ calculated from the viability analysis of Ba/F3 cells carrying indicated compound mutations. Cells were treated with lorlatinib, alectinib, or gilteritinib for 72 h. $N = 3$ independent samples examined over three independent experiments and data presented as mean values ± SD. **d** The suppression of phospho-ALK in Ba/F3 cells carrying single and compound mutations conferring resistance to second-generation or third-generation ALK-TKIs was evaluated via western blotting. Cells were treated with the indicated concentrations of gilteritinib for 3 h ($n = 2$). **e** EML4-ALK I1171N + F1174I expressing JFCR-028-3 cells were subcutaneously transplanted into BALB/c $nu/nu$ mice. When the average tumor volume reached ~200 mm$^3$, the mice were randomized to treatment with vehicle control, alectinib (30 mg/kg), lorlatinib (5 mg/kg), or gilteritinib (30 mg/kg) treatment group once daily for 5 days/week via oral gavage ($n = 6$ per treatment group). At day 41, Alectinib or lorlatinib-treated mice were switched to gilteritinib (30 mg/kg) treatment once daily for 5 days/week. Tumor volumes were measured three times a week. Data are presented as mean values ± SD.

ROS1 and NTRK kinase activity, we assessed the efficacy of gilteritinib against *ROS1* or *NTRK* fusion using cancer cell lines and Ba/F3 models.

*TPM3-NTRK1* fusion gene-positive KM12 colorectal cancer cells were highly sensitive to gilteritinib with an $IC_{50}$ of <30 nM and TPM3-NTRK1 WT Ba/F3 cells were also sensitive to gilteritinib ($IC_{50}$; WT, 13.3 nM). Consistent with the inhibition of cell viability, NTRK1 phosphorylation was also suppressed by 30 nM gilteritinib (Supplementary Fig. 19A–D, and Supplementary Table 3B). To further evaluate the efficacy of gilteritinib, in vivo study was performed using KM12 cells. As well as entrectinib, gilteritinib inhibited the tumor growth and NTRK1 autophosphorylation compared with vehicle control (Supplementary Fig. 19E, F). The NTRK1 G667C and G595R mutations were first identified in patients with entrectinib-resistant tumors. Their lesions were resistant to entrectinib and larotrectinib, whereas ponatinib, foretinib, nintedanib, and cabozantinib could overcome the NTRK1-G667C mutation[39,40]. However, these drugs have not been approved for treating *NTRK1*-positive solid tumors; thus, we further analyzed the sensitivity NTRK1 G667C mutants to gilteritinib. Interestingly, Ba/F3 cells expressing TPM3-NTRK1 G667C were more sensitive to gilteritinib than WT cells ($IC_{50}$; G667C, 12.7 nM) (Supplementary Fig. 20A, B, and Supplementary Table 3B). Western blot analysis demonstrated that gilteritinib, but not entrectinib, inhibited the autophosphorylation of NTRK1 in TPM3-NTRK1 G667C-mutant Ba/F3 cells at a low concentration, but neither inhibitor suppressed NTRK1 autophosphorylation and viability in G595R-mutant cells (Supplementary Fig. 20B–D). Additionally, gilteritinib also displayed potency against *ROS1* fusion gene-expressing lung cancer cell lines (HCC78 cells harbor *SLC34A2-ROS1* and JFCR-168 harbor *CD74-ROS1*; Supplementary Fig. 21A, B) and significant tumor regression was observed in vivo (Supplementary Fig. 21C). Since minimal/no growth inhibition effect was observed in EGFR, KRAS-mutated cancer cells in vitro, to confirm the specific inhibiting activity against these oncogenic driver genes, we generated each EGFR and KRAS overexpressed NIH3T3 cells and gilteritinib was treated in vivo. As expected, subcutaneous tumors of ROS1 or NTRK1 expressing NIH3T3 were shrunk as similar as EML4-ALK-expressing cells, however, KRAS-positive tumors were rapidly increased and EGFR-positive tumors were moderately increased consistent with western blot analysis (Fig. 2d, and Supplementary Fig. 22A–E).

## Discussion

Although multiple ALK-TKIs benefit extending survival, it is inevitable to appear tumor relapse caused by acquired resistance. The third-generation ALK-TKI lorlatinib could overcome second-generation ALK-TKIs-resistant single mutants, however, various EML4-ALK I1171N compound mutations such as I1171N + L1198F, I1171N + L1256F, or I1171N + L1196M can

induce lorlatinib resistance (Supplementary Figs. 23 and 24). In this study, we sought to identify a drug with efficacy against these compound mutants, discovering that gilteritinib (ASP2215) significantly suppressed their growth.

Gilteritinib, multi-kinase inhibitor has been clinically approved for treating FLT3-mutant AML in Japan, Europe, and the US. Thus, using following three individual methods, we verified whether gilteritinib has ALK on-target activity. First, immunoblot analysis demonstrated gilteritinib inhibits ALK autophosphorylation in both EML4-ALK-carrying Ba/F3 cells and ALK-positive NSCLC cells. Additionally, even within low concentration (<30 nM), ALK autophosphorylation of I1171N + F1174I mutant was attenuated by gilteritinib on dependent (Fig. 1b, c and Supplementary Fig. 2). Second, we assessed the biochemical kinase activity of WT, L1196M, and L1198F-mutated ALKs. Then, all kinases showed shift of inhibition curve dependent on ATP concentrations (Fig. 1d and Supplementary Fig. 25). Third, phosphoproteomic analysis demonstrated gilteritinib significantly suppressed phosphorylation at multiple sites of ALK, its adapter proteins, downstream proteins, and EML4 (Fig. 1e, Supplementary Figs. 3, 4 and Supplementary Table 2). Consequently, these results proved gilteritinib has ALK on-target activity.

Subsequently, we revealed that gilteritinib suppressed the growth of single mutants including those in patient-derived cells, and all verified I1171N/S compound mutants at a concentration of <30 nM. Additionally, KARPAS299 cell growth was also inhibited by gilteritinib at a low concentration. In Japan, alectinib is only available ALK-TKI for the treatment of anaplastic large cell lymphoma (ALCL). Thus, our results suggested that gilteritinib treatment might be effective against both ALK-TKI-resistant NSCLC and ALCL.

In an analysis of FLT3-positive AML, among 31 patients who relapsed after gilteritinib treatment, four carried the FLT3-ITD F691L mutation (12.9%). In addition, an in vitro analysis demonstrated that the Y693C/N, G697S, and D698N mutations confer resistance to gilteritinib. From the structural simulation analysis of ALK with gilteritinib, it was clearly revealed that the ALK G1202R and D1203N mutations, which are equivalent to the G697S and D698N mutations in FLT3, conferred resistance to gilteritinib[41]. Thus, our computational simulation might reflect the actual binding mode of the drug.

Interestingly, we found that the EML4-ALK G1202R mutation conferred relative resistance to gilteritinib, whereas the EML4-ALK G1202R + L1198F compound mutant tended to be sensitive to the drug. In addition, the EML4-ALK I1171N + L1198F compound mutant was also more sensitive to gilteritinib than the I1171N single mutant, in line with previous findings that the L1198F mutation increases sensitivity to crizotinib. In our previous report, we observed that the EML4-ALK I1171N + L1256F mutant was re-sensitized to alectinib. Our quantum chemical calculations revealed that the π–π interaction between the

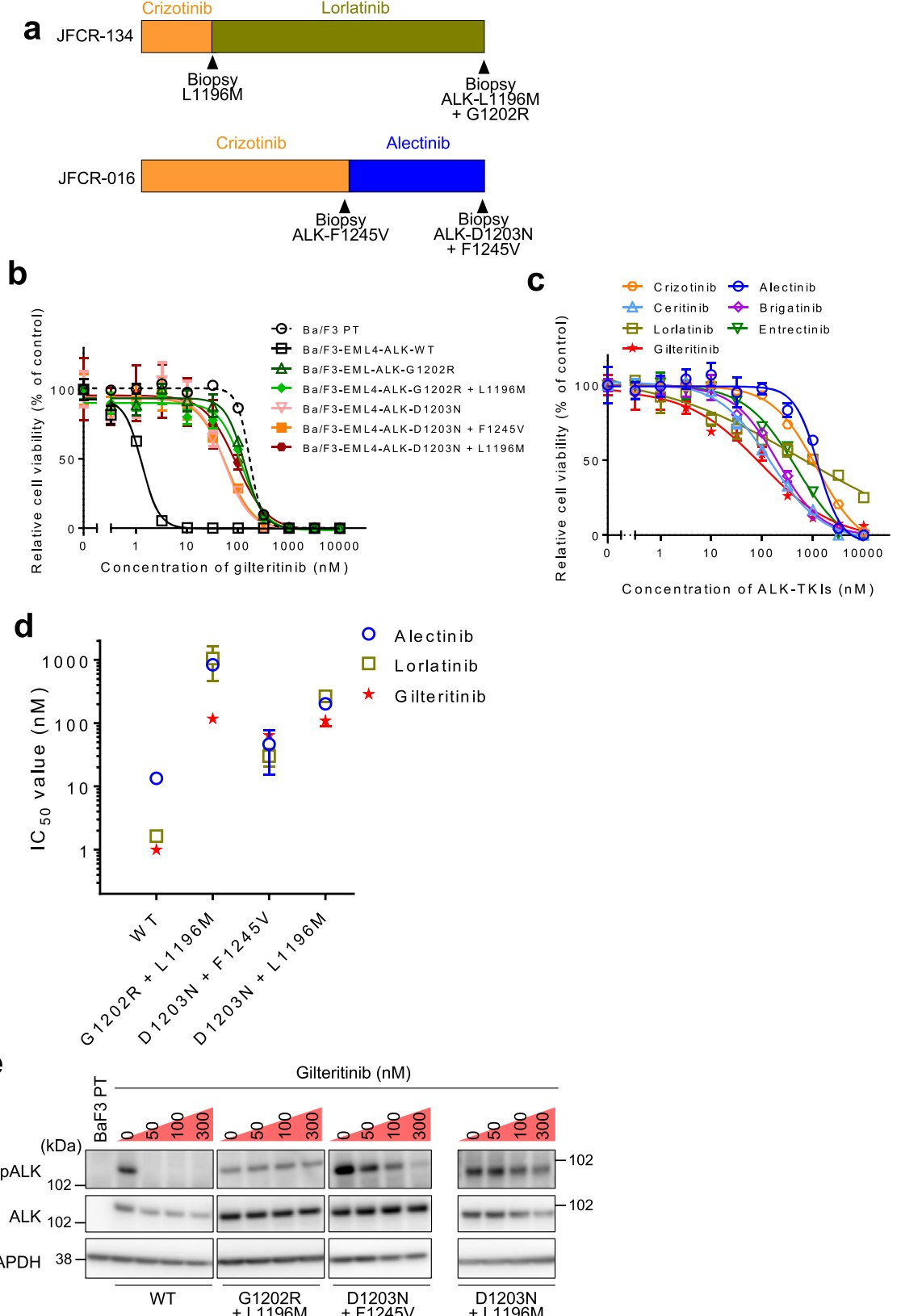

benzocarbazole in alectinib and the phenyl group at ALK-1256F was formed and increased the binding affinity. Our computational simulations in this study suggested the phenyl moiety in gilteritinib is interacted closely with the phenyl group in Phe1198 for the L1198F and G1202R + L1198F mutants (Supplementary Fig. 12B), and the π–π interactions between these aromatic rings might contribute to increasing the binding affinity. The binding mode of gilteritinib with the FLT3 kinase domain based on the

**Fig. 5 Solvent-front mutations and its compound mutants were resistant against gilteritinib. a** Schematic of the timing of ALK-TKI treatment and biopsy for patients from whom the JFCR-134 and JFCR-016. **b** The inhibitory activity of gilteritinib in indicated EML4-ALK expressing Ba/F3 cells. Cells were treated with inhibitors for 72 h and analyzed cell viability using the CellTiter-Glo assay. $N = 3$ independent samples examined over three independent experiments and representative experiment data are presented as mean values ± SD. **c** The inhibitory activity of ALK-TKIs and gilteritinib in MR347 cells. Cells were treated with inhibitors for 72 h and analyzed cell viability using the CellTiter-Glo assay. $N = 3$ independent samples examined over three independent experiments and representative experiment data are presented as mean values ± SD. **d** $IC_{50}$ calculated from the viability analysis of Ba/F3 cells carrying indicated compound mutations. Cells were treated with lorlatinib, alectinib, or gilteritinib for 72 h. $N = 3$ independent samples examined over three independent experiments and data presented as mean values ± SD. **e** The phospho-ALK in Ba/F3 cells carrying G1202R/D1203N compound mutants was evaluated via western blotting. Cells were treated with the indicated concentrations of gilteritinib for 3 h ($n = 2$).

X-ray crystal structure analysis was previously reported, and it illustrated that Y693 is in FLT3, which is analogous to ALK L1198, which is located in the phenyl group of gilteritinib. Although there was no mention on the importance of FLT3 Y693 residue in the binding between gilteritinib and the FLT3 kinase domain, interestingly, amino acid residues harboring an aromatic ring structure, such as tyrosine or phenylalanine might greatly influence the binding affinity between the tyrosine kinase domain and gilteritinib. Indeed, the FLT3-Y693C mutation, which has lost the phenyl ring, induced resistance to gilteritinib[41,42]. To further assess the importance of phenyl ring, we evaluated other amino acid substitution such as L1198I and L1198H. As a result, neither Ile1198 nor His1198 showed more sensitization to gilteritinib (Supplementary Fig. 26). Although histidine also belongs to aromatic amino acids, phenylalanine has a phenyl group and histidine has an imidazole group, respectively. Thus, it might be crucial that when substituted with amino acid we get a phenyl group to increase π–π interaction with gilteritinib. While it is difficult to assess the validity of the clinically effective concentration range of gilteritinib against mutant ALK fusions, our comprehensive data offer important information for treatment selection.

In addition to mutation in the kinase domain, ALK-TKI resistance are also mediated by bypass pathway activation. Mori and colleagues reported that low concentrations of gilteritinib treatment (1 or 5 nM) could suppress ALK, AXL, FLT3, LTK, ROS, or NTRK1 kinase activity in a TK-ELISA or off-chip mobility shift assay[43]. This suggests other RTKs activation could induce resistance against gilteritinib. In this report, we evaluated the effect of KRAS and EGFR activation in ALK-positive NSCLC cells. As expected, the efficacy of gilteritinib was decreased and combination therapy with the respective drugs (KRAS or EGFR inhibitor) markedly inhibited the cell growth in vitro and in vivo (Supplementary Figs. 16–18). Meanwhile, several groups demonstrated that acquired drug resistance may be caused by the presence of drug-tolerance[16] after chemotherapy or molecular target therapy. Concerning the survival of DT cells, AXL is considered a key regulator of DT cells in *EGFR*-mutant NSCLCs[37]. Other groups identified AXL as an ALK-TKI resistance mechanism[44–46]. Then, using AXL expressed H3122 cells, we revealed gilteritinib significantly suppress cell survival and tumor regrowth compared to alectinib (Supplementary Figs. 14 and 15). If AXL kinase plays an important role in DT cell survival in *ALK*-rearranged cancer, the dual inhibition of ALK and AXL by gilteritinib might delay the emergence of additional acquired resistance and represent an important clinical option.

In our study, in vitro and in vivo data demonstrated that gilteritinib potently inhibited WT ROS1. Currently, only crizotinib and entrectinib have received approval for ROS1-positive NSCLC. Thus, our results suggested that gilteritinib treatment might be a new therapeutic option. Of note, we revealed that gilteritinib potently inhibited WT NTRK1 and more potently inhibited G667C-mutant NTRK1. A previous report discovered the NTRK1 G595R and G667C mutations, which are analogous to ALK G1202R and G1269C/A, respectively, in the ctDNA of a patient with entrectinib resistance[47]. Contrarily, G595R-mutant NTRK1 is highly resistant to gilteritinib as well as entrectinib and larotrectinib. NTRK1 G667 and ALK G1269 are located immediately before the DFG motif, a common core motif of the kinase domain that regulates substrate phosphorylation. Therefore, gilteritinib might bind mainly through the hinge region but not adjacent residues of the DFG motif, in contrast to the binding mode between crizotinib and ALK.

In this paper, we discovered that various ALK-TKI-resistant mutants exhibited high sensitivity to gilteritinib. But it is possible that newly acquired resistance to gilteritinib might be also identifiable in the future. According to molecular docking and MD simulations, hydrogen bonds with E1197, M1199, and E1210 residues are important to interact with gilteritinib. Thus, the mutation of these residues suggested from computational analysis, and G1202R and D1203N mutations suggested from our in vitro study (Supplementary Fig. 27 and Supplementary Table 3), could be resistant but further analysis should be elucidated.

In conclusion, the FLT3/AXL inhibitor gilteritinib displayed potent activity against ALK-TKI-resistant EML4-ALK I1171N/S compound mutants and ALK-TKI-resistant single mutants excluding G1202R and D1203N were inhibited by gilteritinib in vitro and in vivo. In addition, gilteritinib inhibited NPM-ALK, ROS1, and NTRK1 kinase activity. Molecular docking and MD simulations revealed that gilteritinib fits into the ATP-binding pocket of ALK, and forms three hydrogen bonds with E1197, M1199, and E1210 residues. Our study represents the first report of the effectiveness of gilteritinib against ALK-TKI resistance EML4-ALK mutants, and these findings might provide beneficial information for the identification of additional indications for gilteritinib.

## Methods

**Cell lines and culture condition**. 293FT human embryonic kidney cells were cultured in Dulbecco's modified Eagle's medium (DMEM) high glucose (Wako, Osaka, Japan) supplemented with 10% fetal bovine serum (FBS) and 250 µg/ml kanamycin (Meiji Seika Pharma, Tokyo, Japan). Ba/F3 immortalized murine bone marrow-derived pro-B cells, NIH3T3 mouse embryonic fibroblast cells, and A431 human epidermoid carcinoma cells were cultured in DMEM low glucose (Wako) supplemented with 10% FBS and kanamycin (D-10) with or without 0.5 ng/ml interleukin (IL)-3 (Invitrogen, Waltham, MA, USA). The human cancer cell lines H2228, H3122, KM12, HCC827, PC9, A549, and H460 were cultured in RPMI1640 (Wako) supplemented with 10% FBS and kanamycin (R-10). The EML4-ALK-positive NSCLC patient-derived cell lines JFCR-018-1 and JFCR-028-3, the alectinib-resistant EML4-ALK-I1171N mutant NSCLC patient-derived cell line MCC-003, the *ROS1* fusion-positive NSCLC patient-derived cell line JFCR-168, the BRAF V600E mutant positive NSCLC patient-derived cell line JFCR-256-3 and the human cancer cell line HCC78 were cultured in medium containing equal proportions of RPMI1640 and Ham's F12 (Wako), and supplemented with 15% FBS and 1× antibiotic–antimycotic mixed stock solution (Wako). The ALK-TKI-resistant NSCLC patient-derived cell line MR347, which carries the EML4-ALK-D1203N + L1196M mutations, was cultured in F-medium [3:1 (v/v) Ham's F12-DMEM, supplemented with 5% FBS, 0.4 µg/ml hydrocortisone (Sigma-Aldrich, St. Louis, MO, USA), 5 µg/ml insulin (Sigma-Aldrich), 8.4 ng/ml cholera toxin (Wako), 10 ng/ml epidermal growth factor (Invitrogen), 24 µg/ml adenine (Sigma-Aldrich), and 10 mM Y-27632 (#A11001-50; AdooQ BioScience, Irvine,

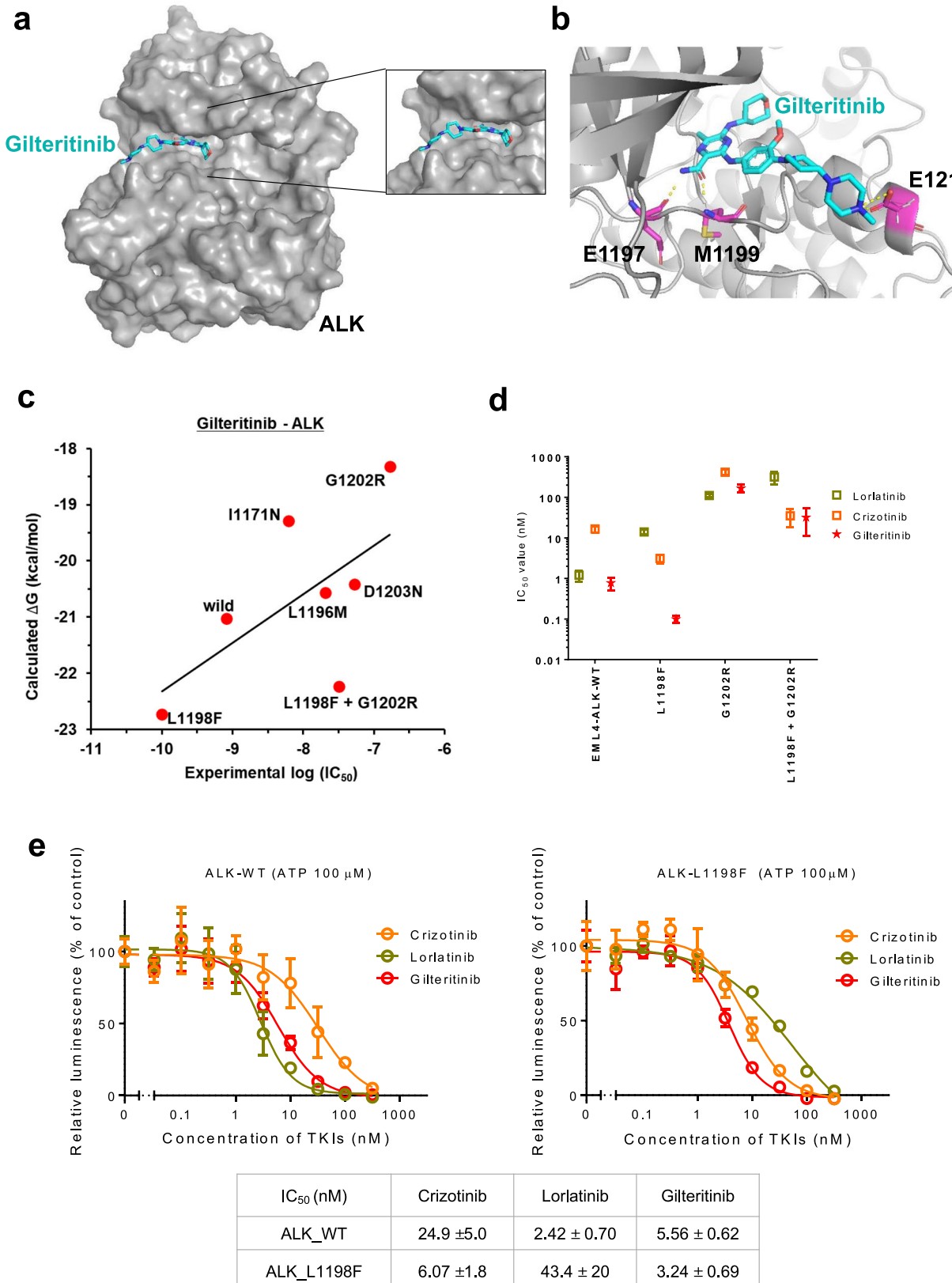

| IC$_{50}$ (nM) | Crizotinib | Lorlatinib | Gilteritinib |
|---|---|---|---|
| ALK_WT | 24.9 ±5.0 | 2.42 ± 0.70 | 5.56 ± 0.62 |
| ALK_L1198F | 6.07 ±1.8 | 43.4 ± 20 | 3.24 ± 0.69 |

CA, USA)] supplemented with 1× antibiotic–antimycotic mixed stock solution. The human non-Hodgkin's Ki-positive large cell lymphoma cell line KARPAS299 was cultured in RPMI1640 supplemented with 20% FBS and 1× penicillin–streptomycin solution (Wako). TIG-3 human lung fibroblasts were cultured in minimum essential medium (Sigma-Aldrich) supplemented with 10% FBS and 1× penicillin–streptomycin solution. The ALK-TKI-resistant NSCLC patient-derived cell line JFCR-098 was cultured in StemPro-hESC+Y medium [1:1 (v/v) Ham's F12-DMEM + GlutaMAX, supplemented with 1× StemPro, 1.6% BSA, 8 ng/ml bFGF, 100 μM 2-Mercapto-ethanol, 10 μM Y-27632] supplemented with 1× antibiotic–antimycotic mixed stock solution.

**Fig. 6 Structure model of the ALK-gilteritinib complex and the predicted binding affinity for ALK mutants. a** The ALK–gilteritinib complex structure predicted by the molecular docking and molecular dynamic (MD) simulation. The mean stable structure of the ALK–gilteritinib complex was extracted from 50 ns × 5 MD simulations, and is represented by surface (ALK) and stick (gilteritinib: C, light blue; N, blue; O, red) models. In the structure model, gilteritinib fits into the ATP-binding pocket in ALK without any steric crushes by overview (left) and zoom-in of the ATP-binding pocket (right). **b** The gilteritinib-binding mode in the ATP-binding pocket in ALK. The protein backbone is represented by a gray ribbon diagram. E1197, M1199, and E1210 are colored with magenta, and their side chains were depicted as sticks (C, magenta; N, blue; O, red), respectively. Hydrogen bonds between these residues and gilteritinib are shown by dashed yellow lines. **c** The binding free energy ($\Delta G$) of gilteritinib to wild-type (WT) or each resistant mutant is plotted against experimental $IC_{50}$ of the corresponding Ba/F3 mutant. These $\Delta G$ values are calculated by MP-CAFEE. The linear association between $\Delta G$ and experimental $IC_{50}$ was calculated by Pearson's product–moment correlation coefficient ($R = 0.627$). **d** $IC_{50}$ calculated from the cell viability assay of WT, L1198F, G1202R and L1198F + G1202R compound mutation-expressing Ba/F3 cells. Cells were treated with crizotinib, lorlatinib, or gilteritinib for 72 h. $N = 3$ independent samples examined over three independent experiments and data presented as mean values ± SD. **e** The evaluation of the sensitization activity of gilteritinib in the in vitro kinase assay using the ADP-Glo assay kit. $IC_{50}$ value calculated at an ATP concentration of 100 μM suggested the better affinity of gilteritinib to ALK-L1198F than to wild-type ALK. $N = 3$ independent samples examined over three independent experiments and representative experiment data are presented as mean values ± SD.

**Establishment of oncogene-expressing cells**. To establish Ba/F3 cells expression WT or mutant EML4-ALK, TPM3-NTRK1, and CD74-ROS1 fusion proteins, pLenti6.3 vectors containing the cDNA of these fusion oncogenes were transfected into 293FT cells using packaging plasmids (ViraPower) for lentivirus production. Ba/F3 cells were infected using lentivirus-containing medium supplemented with polybrene (8 μg/ml), and after 24 h of incubation, the infected cells were selected using 7 μg/ml blasticidin (Invitrogen) for 1 week. After selection, each EML4-ALK-, TPM3-NTRK1-, and CD74-ROS1-expressing cell line was cultured in D-10 without IL-3. FLT3-ITD-expressing Ba/F3 cells, which were provided by Professor Kazuhiro Katayama (Keio University, Tokyo, Japan), were cultured in R-10 medium supplemented with 0.5 μg/ml puromycin (Takara Bio Inc., Shiga, Japan)[48]. To stablise H3122, JFCR-028-3, MCC-003 cells expression WT or mutant EML4-ALK, KRAS G12C, cells were infected using lentivirus-containing medium supplemented with polybrene (8 μg/ml). After 24 h of incubation, the infected cells were selected using 7 μg/ml blasticidin for H3122, JFCR028-3, and 10 μg/ml for MCC-003, respectively. To stablise H3122 cells expression AXL, pHAGE-AXL vector (purchased from addgene) was transfected into 293FT cells using packaging plasmids for lentivirus production. H3122 cells were infected using lentivirus-containing medium supplemented with polybrene (8 μg/ml), and after 24 h of incubation, EGFP-positive cells were sorted using FACSMelody (BD Bioscience, San Jose, CA, USA). To stablise NIH3T3 oncogene-expressing cells, cells were infected using lentivirus-containing medium supplemented with polybrene (8 μg/ml), and after 24 h of incubation, the infected cells were selected using 2 μg/ml blasticidin for 1 week.

**Reagents**. Crizotinib (PF-02341066), brigatinib (AP26113), and lorlatinib (PF-06463922) were purchased from Shanghai Biochempartner (Shanghai, Chaina). Gilteritinib (ASP2215) was purchased from Shanghai Biochempartner and Biovision (Milpitas, CA, USA). Alectinib (CH5424802) and ceritinib (LDK378) were purchased from ActiveBiochem (Kowloon, Hong Kong). Entrectinib (RXDX-101) and AMG510 were purchased from Medchem Express (Monmouth Junction, NJ, USA). Afatinib (BIBW2992) was purchased from ChemieTek (Indianapolis, IN, USA). Trametinib (GSK1120212) was purchased from AdooQ BioScience (Irvine, CA, USA). Brigatinib was dissolved in ethanol for in vitro experiments, and the other drugs were dissolved in dimethyl sulfoxide (DMSO).

**Drug screening**. Parental Ba/F3 cells, EML4-ALK WT cells, or cells expressing EML4-ALK I1171N + F1174I or I1171N + L1198H were seeded into 96-well plates and treated with an original panel of inhibitors including DMSO controls prepared in-house. After 72 h of incubation, the cells were incubated with CellTiter-Glo reagent (Promega, Madison, WI, USA) for 10 min, and luminescence was measured using TriStar LB941 (Berthold Technologies, Bad Wildbad, Germany).

**Cell viability assay**. The cells were seeded in triplicate into 96-well plates and treated with serially diluted inhibitors. For Ba/F3 cells, 2000 cells/well were seeded into 96-well plates in triplicate and cultured in medium containing serially diluted drugs for 72 h. For H2228 and HCC78 cells, 2000 cells/well were seeded into ultra-low attachment 96-well plates (Corning, NY, USA) in triplicate and cultured in three-dimensional cell culture medium (Nissan Chemical Corporation, Tokyo, Japan) for 24 h, followed by culture in three-dimensional cell culture medium containing different concentrations of drugs for an additional 72 h. For other cell lines, 2000 cells/well were seeded into 96-well plates in triplicate and cultured for 24 h. Cells were then cultured in medium containing different concentrations of drugs for an additional 72 h. Cells were subsequently incubated with CellTiter-Glo reagent, and luminescence was measured. To analyze the data, GraphPad Prism version 7.0.4 (GraphPad software) was used. $IC_{50}$ was determined using a non-linear regression model with a sigmoidal dose response in GraphPad.

**Antibodies and immunoblotting**. ALK-TKI-treated cells or tumor tissues were lysed using 1× sodium dodecyl sulfate (SDS) lysis buffer containing 0.1 M Tris (pH 7.5), 10% glycerol, and 1% SDS and boiled at 100 °C for 5 min. The protein concentrations were measured using a BCA protein assay kit (Thermo Fischer Scientific, Waltham, MA, USA). The lysate concentrations were adjusted to 1 mg/ml using lysis buffer, and a 20% volume of sample buffer containing 0.65 M Tris (pH 6.8), 20% 2-mercaptoethanol, 10% glycerol, 3% SDS, and 0.01% bromophenol blue was added. Then, 10 μg of each sample were subjected to SDS–polyacrylamide gel electrophoresis and immunoblotted using antibodies against total ALK (#3633, Cell Signaling Technology, Danvers, MA, USA, 1:1,000), phospho-ALK (Y1604; #3341, Cell Signaling Technology, 1:1,000, Y1282/1283; #9687, Cell Signaling Technology, 1:1000), total S6 ribosomal protein (#2217, Cell Signaling Technology, 1:1000), phospho-S6 ribosomal protein (#5364, Cell Signaling Technology, 1:10,000), total p42/44 ERK/MAPK (#9102, Cell Signaling Technology, 1:1000), phospho-p42/44 ERK/MAPK (#9101, Cell Signaling Technology, 1:2000), total AKT (#4691, Cell Signaling Technology, 1:1000), phospho-AKT (#4060, Cell Signaling Technology, 1:1000), total EGFR (#4267, Cell Signaling Technology, 1:1000), phospho-EGFR (#ab5644, Abcam, Cambridge, UK, 1:1000), total MEK1/2 (#9122, Cell Signaling Technology, 1:1000), phospho-MEK1/2 (#9121, Cell Signaling Technology, 1:1000), PARP (#9542, Cell Signaling Technology, 1:1000), total AXL (#4566, Cell Signaling Technology, 1:1000), phospho-AXL (#5724, Cell Signaling Technology, 1:500), KRAS (#WH0003845M1,Sigma-Aldrich, 1:1000), total NTRK1 (#4609, Cell Signaling Technology, 1:1000), phospho-NTRK1 (#4621, Cell Signaling Technology, 1:1000), total STAT3 (#4904, Cell Signaling Technology, 1:1000), phospho-STAT3 (#9145, Cell Signaling Technology, 1:1000), and GAPDH (MAB374, Millipore, Burlington, MA, USA, 1:1000).

**In vitro kinase assay of ALK protein and ALK inhibitors**. The recombinant proteins of the kinase domain of WT, F1174L, L1196M, L1198F, G1202R, G1269A were purchased from Signal Chem (Richmond, Canada). Appropriate amounts of target proteins were calculated as recommended by the ADP-Glo assay manufacturer's protocol after incubating in 96-well half-area white plates with serially diluted inhibitor over a 9-dose range for 15 min at the room temperature. ATP at concentrations of 1, 10, 100, and 1000 μM was mixed with 200 μg ml⁻¹ substrate and added to a kinase protein–inhibitor mixture, followed by incubation for 60 min at the room temperature. After the kinase reaction, an equal volume of ADP-Glo Reagent was added to terminate the kinase reaction, and the resultant ADP level was measured according to the manufacturer's instructions. The light generated by the luciferase/luciferin reaction was measured using the TriStar LB941 Luminometer.

**Preparation of samples for global phosphoproteomics**. After cells were treated with DMSO or gilteritinib for 3 h, lysis of the cell pellets was performed in lysis buffer (50 mM NaHCO₃, 12 mM Sodium N-Lauroylsarcosinate, 12 mM sodium deoxycholate) supplemented with cOmplete EDTA-free and PhosSTOP (Roche, Basel, Switzerland).

**Phosphoproteome analysis**. Each sample was boiled at 95 °C for 5 min. The lysates were further sonicated with a Bioruptor sonicator (Cosmo Bio, Tokyo, Japan). Then, the samples were reduced with 10 mM TCEP, alkylated with 20 mM iodoacetamide, and quenched with 21 mM of L-cysteine. Samples were digested with trypsin (protein weight: 1/50) and Lys-C (protein weight: 1/50) for 16 h at 37 °C. Samples were acidified with 1% TFA and centrifuged at 20,000×g for 10 min at 4 C. Supernatants were desalted and applied IMAC column for phosphopeptide enrichment[49]. TMTpro 16plex reagents (0.5 mg, Thermo Scientific, Bremen, Germany) were dissolved in anhydrous acetonitrile (40 μL) of which 4 μL was added to the phosphopeptides desolved with 10 μL of 100 mM TEAB. Following incubation at room temperature for 1 h, the reaction was quenched with hydroxylamine to a final concentration of 0.3% (v/v). The TMT-labeled samples were

pooled and divided into 10% for global phosphoproteome analysis and 90% for phosphotyrosine (pY) proteome analysis. Both samples were vacuum centrifuged to near dryness.

For global phosphoproteome analysis, TMT-labeled phosphopeptides were subjected to off-line basic pH reversed-phase (BPRP) fractionation. We used Thermo Scientific UltiMate 3000 UHPLC system equipped with a dual wavelength detector (set at 220 and 280 nm). Mobile phases were BPRP-A (5 mM ammonium bicarbonate pH 9.2) and BPRP-B (5 mM ammonium bicarbonate pH 9.2 and 60% acetonitrile). The LC gradient was ramped up 1–19.5% BPRP-B for 7 min and then 19.5–64% BPRP-B for 27 min. Peptides were separated by L-column3 C18 column (5 μm particles, 0.3 mm ID and 150 mm in length, Chemicals Evaluation and Research Institute, Tokyo, Japan) at the flow rate of 2 μL/min. The peptide mixture was fractionated into a total of 21 fractions which were consolidated into 7 fractions. Samples were subsequently vacuum centrifuged to near dryness. Each fraction was reconstituted in 2% acetonitrile, 0.1% trifluoroacetic acid for LC–MS/MS processing.

For pY proteome analysis, pY peptides were dissolved in IP buffer (5 mM Tris–HCl (pH 7.5), 5 mM NaCl) and enriched using pY1000 antibody-conjugated Dynabeads[50]. pY peptides on the Dynabeads were eluted two times with 60% acetonitrile (ACN)/0.1% TFA. For depletion of pY1000 antibodies, the eluted pY peptides were applied to $Fe^{3+}$ IMAC columns on a C18 disc in a stop-and-go extraction tip. After drying the elutes, the pY peptides were re-suspended in 10 μl of 2% ACN/1% TFA for LC–MS/MS analysis.

LC–MS/MS was performed by coupling an UltiMate 3000 Nano LC system (Thermo Scientific) and an HTC-PAL autosampler (CTC Analytics, Zwingen, Switzerland) to an Orbitrap Fusion Lumos mass spectrometer (Thermo Scientific). Peptides were delivered to an analytical column (75 μm × 30 cm, packed in-house with ReproSil-Pur C18-AQ, 1.9 μm resin, Dr. Maisch, Ammerbuch, Germany) and separated at a flow rate of 280 nL/min using a 145-min gradient from 5% to 30% of solvent B (solvent A, 0.1% FA; solvent B, 0.1% FA and 99.9% acetonitrile). The Orbitrap Fusion Lumos mass spectrometer was operated in the data-dependent mode. For global phosphoproteome analysis, survey full scan MS spectra ($m/z$ 375–1500) were acquired in the Orbitrap with 120,000 resolution after accumulation of ions to a $4 \times 10^5$ target value. Maximum injection time was set to 50 ms and dynamic exclusion was set to 30 s. MS2 analysis consisted of higher-energy collisional dissociation; AGC $1 \times 10^5$; normalized collision energy 38; maximum injection time 105 ms; 50,000 resolution and isolation window of 0.7 Da. For pY proteome analysis, survey full scan MS spectra ($m/z$ 375–1500) were acquired in the Orbitrap with 120,000 resolution after accumulation of ions to a $1 \times 10^5$ target value. Maximum injection time was set to 100 ms and dynamic exclusion was set to 10 s. MS2 analysis consisted of higher-energy collisional dissociation; AGC $1 \times 10^5$; normalized collision energy 32; maximum injection time 315 ms; 120,000 resolution and isolation window of 0.7 Da. Raw MS data were processed by MaxQuant (version 1.6.14.0) supported by the Andromeda search engine. The MS/MS spectra were searched against the UniProt human database with the following search parameters: full tryptic specificity, up to two missed cleavage sites, carbamidomethylation of cysteine residues set as a fixed modification, and serine, threonine, and tyrosine phosphorylation, N-terminal protein acetylation and methionine oxidation as variable modifications. The false discovery rate of protein groups, peptides, and phosphosites were <0.01. Quantitative values of the phosphorylation sites across different fractions were automatically integrated and summarized in "Phospho (STY) Sites.txt" by MaxQuant.

**Apoptosis assay.** H3122, KM12, or MCC-003 cells ($1 \times 10^5$) were seeded into six-well plates and cultured in appropriate medium. After 24 h, cells were cultured in drug-containing medium (100 nM) for an additional 72 h. All floating and adherent cells were collected and stained with Alexa Fluor 647-labeled Annexin-V and propidium iodide using a Dead Cell Apoptosis Kit (Thermo Fischer Scientific) for 15 min at room temperature in the dark. Each sample was evaluated using FACSVerse (BD Bioscience). Output data were analyzed using FlowJo software (BD, Franklin Lakes, NJ, USA).

**In vivo study of gilteritinib and TKIs.** All in vivo studies were conducted in line with the protocols approved by the Committee for the Use and Care of experimental animals of the Japanese Foundation for Cancer Research. Mice were housed in a controlled-temperature room maintained on a 12 h light/dark cycle. In total, $2.5 \times 10^6$ cells for H3122, JFCR-028-3, and MCC-003, $3 \times 10^6$ cells for KM12, $5 \times 10^6$ cells for NIH3T3 were subcutaneously injected into BALB/c nu/nu mice (Charles River, Wilmington, MA, USA). $3 \times 10^6$ cells for JFCR-168 were subcutaneously injected into SCID Beige mice (Charles River). After the tumor volumes reached ~200 mm³, the mice were orally administered the targeted drugs. Tumor volume and body weight were measured three times a week to evaluate the tumor growth rate, which was calculated using the following formula: 0.5 × length × width². When the tumor size exceeded 1000 mm³, the mice were euthanized. Statistical P values were calculated by two-sided Mann–Whitney U test using Graphpad Prism software, Version 7.0.4, significant P values are shown as *P < 0.05, **P < 0.01.

**Molecular docking.** Molecular docking of gilteritinib toward the ALK-tyrosine kinase domain was performed using the genetic algorithm-docking software GOLD 5.5. The standard default settings for the genetic algorithm were used. The initial structural data of the protein was obtained from the crystal structure of the human ALK complex with brigatinib (PDBID: 6MX8), and the structures of disordered loops and flexible side chains were modeled using the structure preparation module in the Molecular Operating Environment (MOE) (v. 2016.08 (Chemical Computing Group Inc., Montreal, Canada). The N- and C-termini of the protein model were capped with acetyl and N-methyl groups, respectively. The dominant protonation state at pH 7.0 was assigned for titratable residues. The ATP-binding site in ALK was defined to include all atoms within 10A of the midpoint of Leu1122 Ca and Gly1202 Ca atoms. Gilteritinib, whose 3D structure was obtained from the crystal structure of FLT3 complexed with gilteritinib (PDBID: 6JQR), was protonated to form ionization states in solution (the net charge of +2). After the backbone Ca atoms in ALK were structurally aligned with those in the FLT3–gilteritinib complex, gilteritinib was docked into the ATP-binding site in ALK with positional restraint on the pyrazinamide moiety, assuming that gilteritinib has a similar binding geometry between ALK and FLT3. The top-ranked docking pose was extracted and used as the initial structure of following MD simulations of the ALK–gilteritinib complex.

**MD simulation of WT ALK or its mutants in complex with gilteritinib.** Each of I1171N, L1196M, L1198F, G1202R, D1203N or G1202R + L1198F mutations was introduced into the structure of WT ALK using the structure preparation module in MOE. Gilteritinib structure was optimized, and the electrostatic potential was calculated at the HF/6-31G* level using the GAMESS program, after which the atomic partial charges were obtained by the RESP approach[51]. The other parameters for the compound were determined by the general Amber force field using the antechamber module of AMBER Tools 12[51]. The Amber ff99SB-ILDN force field was used for protein and ions and TIP3P was used for water molecules[51]. Water molecules were placed around the complex model with an encompassing distance of 8A, including roughly 13,000 water molecules. Charge-neutralizing ions were added to neutralize the system. All MD simulations were carried out using the GROMACS 4 program on high performance computing infrastructure (HPCI)[51]. We used the same simulation parameters as described by Uchibori et al.[51]. After the fully solvated system was energy-minimized, it was equilibrated for 100 ps under a constant number of molecules, volume, and temperature condition, and run for 100 ps under a constant number of molecules, pressure, and temperature (NPT) condition, with positional restraints on the protein's heavy atoms and compound atoms. Each production run was conducted under the NPT condition without the positional restraints. For each ALK mutant, five sets of 50 ns production runs were performed with different velocities. Three sets of 20 ns production runs were performed for the solvated gilteritinib system.

The binding free energy (ΔG) between ALK and gilteritinib was calculated by MP-CAFEE (massively parallel computation of absolute binding free energy with well-equilibrated states), which is one of the alchemical free energy perturbation methods[52]. ΔG for each ALK mutant was computed according to a protocol described in the previous study[53].

**Reporting summary.** Further information on research design is available in the Nature Research Reporting Summary linked to this article.

## Data availability

The phosphoproteomics data used in this manuscript have been deposited to the jPOSTrepo with the identifier JPST001063. All the other data supporting the findings of this study are available within the article and its Supplementary Information files and from the corresponding authors upon reasonable request. A reporting summary is available as a Supplementary Information file. Source data are provided with this paper.

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

## Acknowledgements

This study was supported in part by MEXT/JSPS KAKENHI grant number 17H06327 (to N.F.), 19H03524 and 20K21554 (to R.K.), 18K06594 (to M.A.), the grant from the AMED grant number JP20cm0106203h0005 and JP20ck0106472h0002 (to R.K.), and the grant from Nippon Foundation (to N.F.), and Uehara Memorial Foundation (to R.K.). MEXT as "Program for Promoting Researches on the Supercomputer Fugaku (Application of Molecular Dynamics Simulation to Precision Medicine Using Big Data Integration System for Drug Discovery)" (to Y.O.), and FOCUS Establishing Supercomputing Center of Excellence (to Y.O.). This research used computational resources of the HPCI system provided by Information Technology Center, the University of Tokyo (Oakbridge-CX) through the HPCI System Research Project (Project ID: hp200129). J.K. received a scholarship from Polish National Agency for Academic Exchange.

## Author contributions

H.M. and K.O. designed the experiments, performed cell line and in vivo studies, and wrote the manuscript. J.A. performed phosphoproteome analysis. B.M., Y. Sasakura, and Y. Sagae performed computational simulations, and M.A. and Y.O. supervised the simulations, and wrote the manuscript. J.K. and T.O. designed the experiments, performed cell line and in vitro studies. A.T. performed in vivo studies, M.K.-N. and M.S. performed protein purification and crystallization. N.Y., K.W., and M.N identified

patients and obtained repeat biopsy samples, and analyzed clinical data. L.F. and K.T. established the patient-derived cells. K.K. established Ba/F3-FLT3-ITD cells. S.T., S.S., and N.F. supervised the in vitro and in vivo experiments, and wrote the manuscript. R.K. designed the study, performed cell line, patient-derived cells, and in vitro and in vivo experiments, supervised the experiments, and wrote the manuscript.

## Competing interests

R.K. received research grants from Chugai, TAKEDA, Toppan Printing, Daiichi-Sankyo. N.F. received research grants from API corporation. The remaining authors declare no competing interests.

## Additional information

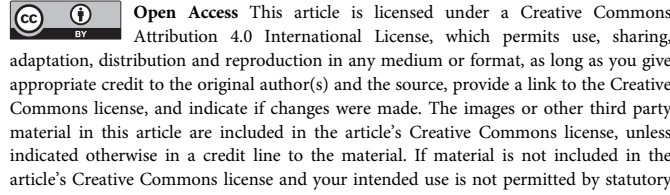

