## [Peer Review File · Nature Communications]

Reviewers' Comments:

Reviewer #1:

Remarks to the Author:

The authors demonstrate the efficacy of gilteritinib, an approved drug for the treatment of FLT3+ AML, in various cellular expressing ALK, ROS1 and NTRK1 rearrangements in combination with different known drug resistance mutations. Gilteritinib seems to overcome various ALK resistance mutations against ALK specific TKIs. The major limitation of the study is a complete lack of characterization of potential off-targets for gilteritinib since this is a multi-kinase inhibitor hitting a wide range of tyrosine kinases and potentially even other enzymes. The authors also provide a very limited dataset to explain the exquisite efficacy of gilteritinib against the different mutants. Overall, this is an interesting dataset but a lack of overall clarity of the manuscript and novelty of the data limits the enthusiasm for the paper.

Major

- The authors need to present data (e.g. immunoblotting, phosphoproteome, etc.) to control the effects of gilteritinib on at least the most prominent targets of gilteritinib beyond the driver oncogene itself. Without that data it remains unclear to which extent the efficacy of the compound is driven by on-target activity
- The authors need to provide a more thorough explanation than just a docking model of gilteritinib to explain the efficacy of the compound. This could be done either by an actual co-crystal structure of gilteritinib in ALK or a related kinase. Alternatively, the authors could use biochemical assays of the isolated mutants to test assess the kinetics of gilteritinib activity in the different mutants. Here, especially the hypersensitization of the ALK kinase through L1198F could be explored more broadly with the mutagenesis of the Phenylalanine to another AA.
- The authors present in vivo data of xenograft that are not relevant for the on-target activity of gilteritinib against the various resistance mutants. The impact of the study would increase dramatically if the authors could provide that data in one of the relevant resistance mutations such as I1171N+F1174I or any related mutants.
- The structure of the manuscript is really confusing. The authors start off with double mutants of ALK, go via single mutants of ALK to in vivo studies, later more compound mutations to end the manuscript with the activity of gilteritinib in NTRK rearranged cells. This very much limits the clarity of the findings and it remains elusive in what way the different data points are related to each other.

Minor

- Typing errors and missing words (e.g. line 322; 330)
- Annotations of WB (5B non=DMSO?)
- Color coding is not very helpful to distinguish the different data points

Reviewer #2:

Remarks to the Author:

In this manuscript, the authors report that the multi-kinase inhibitor gilteritinib is effective against on-target mutations that arise during ALK inhibitor treatment in ALK fusion positive lung cancer to cause resistance, particularly to advanced generations of ALK kinase inhibitors. The findings arose out of a focused inhibitor screen in genetically defined isogenic systems and were confirmed in patient-derived models. Gilteritinib was also effective against ROS1 and NTRK fusion positive tumors.

The manuscript is interesting and the findings are solid. There are issues to address to improve the strength of the study.

(1) The authors should conduct in vivo studies using ROS1 and NTRK fusion patient derived models to increase the translational impact.

(2) Some important mechanisms of resistance to ALK inhibitors are not mentioned, such as lineage switching and activation of the MAPK pathway (PMIDs: 22235099, 26301689). These works should be mentioned and referenced. Further, it should be tested whether gilteritinib can overcome resistance that is induced by KRAS upregulation or mutation in the patient derived models.

(3) Since gilteritinib is a multi-kinase inhibitor, which the authors acknowledge, experiments should be conducted to test whether upregulation of AXL or the other relevant kinase targets of gilteritinib mentioned by the authors cause resistance to gilteritinib or not. This will help anticipate the potential clinical utility in the future.

(4) Is there a rationale for combining gilteritinib with other current ALK TKIs in clinical use to prevent or overcome resistance?

(5) The grammar of the text could be improved.

Rebuttal Letter for NCOMMs-20-19430R
Point-by point authors' answers to the editorial comments

Reviewer #1 (Remarks to the Author):

The authors demonstrate the efficacy of gilteritinib, an approved drug for the treatment of FLT3+ AML, in various cellular expressing ALK, ROS1 and NTRK1 rearrangements in combination with different known drug resistance mutations. Gilteritinib seems to overcome various ALK resistance mutations against ALK specific TKIs. The major limitation of the study is a complete lack of characterization of potential off-targets for gilteritinib since this is a multi-kinase inhibitor hitting a wide range of tyrosine kinases and potentially even other enzymes. The authors also provide a very limited dataset to explain the exquisite efficacy of gilteritinib against the different mutants. Overall, this is an interesting dataset but a lack of overall clarity of the manuscript and novelty of the data limits the enthusiasm for the paper.

-> Thank you very much for reviewing our manuscript. According to the comments raised by the reviewers, we performed a number of experiments including multiple *in vivo* experiments and added new data to solidify our findings and strengthen the significance of this manuscript. I hope the revised manuscript is worth to share with researchers by publishing in Nature Communications.

Major

- The authors need to present data (e.g. immunoblotting, phosphoproteome, etc.) to control the effects of gilteritinib on at least the most prominent targets of gilteritinib beyond the driver oncogene itself. Without that data it remains unclear to which extent the efficacy of the compound is driven by on-target activity

-> Thank you very much for your important comments. To answer this comment, we performed phospho-proteomics analysis using ALK-rearranged NSCLC patient derived cells by treating gilteritinib. As the results, gilteritinib significantly decreased phosphorylation of ALK and its adaptor proteins such as IRS1/2, SOS2 or SH2B2. Further, using the phospho-proteomics data, kinase substrate enrichment analysis (KSEA) predicted activate kinase candidates as potential gilteritinib targets and identified ALK as the most likely regulated protein. We added the data in Fig. 1E, Supplemental Figs. 3-4 and Supplemental Table 2.

We also performed immunoblot analysis to check whether gilteritinib inhibit not only ALK but also its down stream molecules. As the results, gilteritinib inhibited ALK auto-phosphorylation and its downstream signals at the same concentration in several ALK rearranged cells (Fig 1C, 2D and Supplemental Fig 2).

- The authors need to provide a more thorough explanation than just a docking model of gilteritinib to explain the efficacy of the compound. This could be done either by an actual co-crystal structure of gilteritinib in ALK or a related kinase. Alternatively, the authors could use biochemical assays of the isolated mutants to test assess the kinetics of gilteritinib activity in the different mutants. Here, especially the hypersensitization of the ALK kinase through L1198F could be explored more broadly with the mutagenesis of the Phenylalanine to another AA.

-> Thank you very much for your import comments. To answer this comment, we conducted in vitro kinase assay to clearly demonstrate whether gilteritinib directly inhibit ALK tyrosine kinase. In vitro kinase assay using purified ALK protein (kinase domain of WT, L1198F, L1196M) showed that gilteritinib inhibited ALK kinase activity dose dependent manner, and IC50 was shifted by the increasing concentration of ATP (suggesting gilteritinib inhibited ALK tyrosine kinase ATP competitive manner). In addition, ALK-L1198F kinase was more sensitive to gilteritinib (and crizotinib but not lorlatinib) than WT-ALK. The data was added in Fig. 1D, 6E and Supplemental Fig. 23.

We also tried to obtain actual crystal structure, and we could get the crystal of ALK kinase, but the

size is still too small to get the Xray diffraction data (please see below). Now we're thinking that we'll continue to get the crystal structure data for our future study.

- The authors present in vivo data of xenograft that are not relevant for the on-target activity of gilteritinib against the various resistance mutants. The impact of the study would increase dramatically if the authors could provide that data in one of the relevant resistance mutations such as I1171N+F1174I or any related mutants.

-> Thank you very much for your important comments. According to the comments, we performed in vivo experiments using EML4-ALK-I1171N+F1174I expressed EML4-ALK positive NSCLC cells. As shown in Fig 4E, I1171N+F1174I expressed tumor were not completely shrunk but regrew within short periods on alectinib or lorlatinib treatment. In contrast, gilteritinib induced complete remission of tumor over 50 days. In addition, after the tumor regrowth on lorlatinib or alectinib treatment, gilteritinib were administered to those alectinib or lorlatinib resistant tumor bearing mice. To our surprise, the immediate tumor shrinkage was observed by switching to gilteritinib treatment.

- The structure of the manuscript is really confusing. The authors start off with double mutants of ALK, go via single mutants of ALK to in in vivo studies, later more compound mutations to end the manuscript with the activity of gilteritinib in NTRK rearranged cells. This very much limits the clarity of the findings and it remains elusive in what way the different data points are related to each other.

-> Thank you very much for your important comments. We totally agree with your comments. We largely changed the structure of the manuscript.

Minor

- Typing errors and missing words (e.g. line 322; 330)

-> I am sorry for the typing error. We corrected those errors and the whole manuscript was checked by the English editing service company "Enago (Crimson Interactive Pvt. Ltd)".

- Annotations of WB (5B non=DMSO?)

-> Thank you for the comments. We adequately corrected (Supplementary Fig 17A).

- Color coding is not very helpful to distinguish the different data points

-> I am sorry for our color coding. We totally changed the color coding.

Reviewer #2 (Remarks to the Author):

In this manuscript, the authors report that the multi-kinase inhibitor gilteritinib is effective against on-target mutations that arise during ALK inhibitor treatment in ALK fusion positive lung cancer to cause resistance, particularly to advanced generations of ALK kinase inhibitors. The findings arose out of a focused inhibitor screen in genetically defined isogenic systems and were confirmed in patient-derived models. Gilteritinib was also effective against ROS1 and NTRK fusion positive tumors.

The manuscript is interesting and the findings are solid. There are issues to address to improve the strength of the study.

-> Thank you very much for reviewing our manuscript, and thank you very much for your encouraging comments. According to the following comments by the reviewer, we performed a number of experiments including multiple *in vivo* experiments and added new data to solidify our findings and strengthen the significance of this manuscript. I hope the revised manuscript is worth to share with researchers by publishing in Nature Communications.

(1) The authors should conduct *in vivo* studies using ROS1 and NTRK fusion patient derived models to increase the translational impact.

-> Thank you very much for your important comments. According to the comments, we performed *in vivo* experiments using NTRK1 fusion positive cancer cell xenograft, and ROS1 fusion positive lung cancer patient derived cell line xenograft models. As the results, gilteritinib showed marked suppression of tumor growth in both NTRK1 and ROS1 rearranged cancer *in vivo*. We added the data in Fig 8B and Supplementary figure 18C

(2) Some important mechanisms of resistance to ALK inhibitors are not mentioned, such as lineage switching and activation of the MAPK pathway (PMIDs: 22235099, 26301689). These works should be mentioned and referenced. Further, it should be tested whether gilteritinib can overcome resistance that is induced by KRAS upregulation or mutation in the patient derived models.

-> Thank you very much for your important comments. We added the description of the lineage switching and activation of the MAPK pathway and cited these papers (PMIDs: 22235099,

26301689). In addition, we introduced mutant KRAS (G12C) in 2 ALK positive NSCLC patient derived cell lines, and examined the efficacy of gilteritinib in vitro and in vivo. As the results, combination of KRAS inhibitor (AMG510) with gilteritinib completely induced tumor shrinkage, but gilteritinib monotherapy partially suppressed tumor growth. We added those data in Figure 7D-F and Supplementary figure 15.

(3) Since gilteritinib is a multi-kinase inhibitor, which the authors acknowledge, experiments should be conducted to test whether upregulation of AXL or the other relevant kinase targets of gilteritinib mentioned by the authors cause resistance to gilteritinib or not. This will help anticipate the potential clinical utility in the future.

-> Thank you very much for very important and interesting comment. We introduced Axl to ALK positive H3122 cells and examined the efficacy of alectinib and gilteritinib in vitro and in vivo. From the in vivo experiments, Alectinib treatment showed partial tumor growth suppression, but about after 3 weeks, tumor regrowth was observed. As alectinib treated parental H3122 tumors were not regrowth during the same period, it was suggested that observed resistance was dependent on the overexpressed AXL. On the other hand, gilteritinib significantly suppressed tumor growth and the tumor shrinkage was maintained over 5 weeks. Moreover, to evaluate the efficacy of gilteritinib against the tumors that showed resistant to alectinib, the alectinib treated mice were randomized into 2 groups and gilteritinib was administered in one group, and the other group was continued alectinib administration. As a result, gilteritinib treatment clearly inhibited the tumor growth compared with the continuous alectinib treatment. We added the data in Figure 7. In addition, we evaluated the efficacy of gilteritinib on EGFR activation mediated resistant model (patient derived JFCR-098 cells). Since gilteritinib can not effectively inhibit EGFR, gilteritinib with EGFR inhibitor afatinib, but not gilteritinib monotherapy effectively inhibit the growth of JFCR-098 cells (Supplementary Figure S16)

(4) Is there a rationale for combining gilteritinib with other current ALK TKIs in clinical use to prevent or overcome resistance?

-> Thank you for very much for very interesting comments. We performed the following experiments to examine the efficacy of ALK-TKI combination therapy. Using heterogeneous ALK-TKI resistant population model that was derived from two individual resistant mutations, G1202R and I1171N + L1256F, we evaluated the proportion of each mutant expressed H3122 cells and cell proliferation (Following Fig. A, below). As expected, each single treatment induced the dominant population of I1171N + L1256F expressed cells (after lorlatinib treatment) and G1202R expressed cells (after gilteritinib treatment), respectively. Using this heterogeneous resistant model, we assessed the relative cell viability between single treatment and combination treatment with half concentration of lorlatinib and gilteritinib. Compared with each single treatment, combination of half-dose drug treatment significantly inhibited cell growth (Following Fig. B, below). These findings suggested that ALK-TKIs combination therapy with gilteritinib has a potential to overcome and prevent the heterogeneous ALK mutated resistant tumor. In addition, as indicated in the revised manuscript, current approved ALK-TKI with gilteritinib prevent the emergence of AXL mediated resistant tumor, since gilteritinib is a potent AXL inhibitor. However, in a future study, it is needed to elucidate the best combination strategy to prevent the emergence of drug tolerant cells and acquired resistance by inhibiting other bypass pathway essential for the survival of drug tolerant or acquired resistant cells.

(5) The grammar of the text could be improved.

-> I am sorry for the grammatical error. We corrected those errors and the whole manuscript was

double checked by the English editing service company “Enago (Crimson Interactive Pvt. Ltd)”.

The efficacy of combination therapy with gilteritinib and lorlatinib

(A) Graphic depicting the scheme to evaluate the combination therapy with gilteritinib and lorlatinib. (B) Relative cell growth of EML4-ALK G1202R expressed H3122 and I1171N + L1256F expressed H3122 cells. The same number of G1202R or I1171N + L1256F expressed H3122 cells were mixed and seeded into 96-well plates in triplicate. After 24 h, cells were cultured in medium containing indicated concentrations of drugs for an additional 96 h. Cells were subsequently incubated with CellTiter-Glo reagent, and luminescence was measured.

Reviewers' Comments:

Reviewer #1:

Remarks to the Author:

The authors have addressed my point properly with regard to the potential off-target effects of gilteritinib. The figure displayed in 1E would benefit from a legend and it also might be helpful to annotate the hits beyond a certain threshold. Also, the addition of the biochemical assays further strengthens the manuscript.

The same is true for the addition of the in vivo data for gilteritinib after outgrowth of tumors during alectinib or lorlatinib treatment.

I am still not convinced that the structure of the manuscript is optimal since the majority of the data deals with the impact of gilteritinib on different ALK mutations. Therefore it remains unclear to me why the authors choose to end the manuscript with Fig 7/8. In Fig. 7 the authors claim that gilteritinib might be useful in the context of activation of bypass mechanisms such as AXL or KRAS. Several open questions remain here and make this point rather a weak spot and in my view do not qualify for a good punch line. This is also true for the NTRK data in Fig. 8. It starts with the notion that entrectinib is more effective than gilteritinib as shown in 8a but then suddenly in 8b it is the other way round in vivo. Are the authors sure that the dosing presented in 8b are correct to compare the effects of those 2 drugs? The data presented in 8c is not very helpful to make this decision. Thus, presenting the data like this distorts the efficacy claim made for gilteritinib and again overall this data rather weakens the enthusiasm for the solid manuscript presented until Fig. 6. With all due respect to the authors and all the humility necessary since they need to make the call - my suggestion would be to place both Fig 7/8 into the supplement.

Minor

Typos and grammar remain major issues:

e.g. Sentence in lines 299-301: "Thus, in this study, we focused on whether...."

e.g. 310-312:" In vivo analysis consistently demonstrated whereas both...."

Those are just the few that I picked.

Reviewer #2:

Remarks to the Author:

The authors have addressed my comments and the revised manuscript is improved.

Rebuttal Letter for NCOMMs-20-19430R
Point-by point authors' answers to the editorial comments

Reviewer #1 (Remarks to the Author):

The authors have addressed my point properly with regard to the potential off-target effects of gilteritinib. The figure displayed in 1E would benefit from a legend and it also might be helpful to annotate the hits beyond a certain threshold. Also, the addition of the biochemical assays further strengthens the manuscript.

The same is true for the addition of the in vivo data for gilteritinib after outgrowth of tumors during alectinib or lorlatinib treatment.

I am still not convinced that the structure of the manuscript is optimal since the majority of the data deals with the impact of gilteritinib on different ALK mutations. Therefore it remains unclear to me why the authors choose to end the manuscript with Fig 7/8. In Fig. 7 the authors claim that gilteritinib might be useful in the context of activation of bypass mechanisms such as AXL or KRAS. Several open questions remain here and make this point rather a weak spot and in my view do not qualify for a good punch line. This is also true for the NTRK data in Fig. 8. It starts with the notion that entrectinib is more effective than gilteritinib as shown in 8a but then suddenly in 8b it is the other way round in vivo. Are the authors sure that the dosing presented in 8b are correct to compare the effects of those 2 drugs? The data presented in 8c is not very helpful to make this decision. Thus, presenting the data like this distorts the efficacy claim made for gilteritinib and again overall this data rather weakens the enthusiasm for the solid manuscript presented until Fig. 6. With all due respect to the authors and all the humility necessary since they need to make the call - my suggestion would be to place both Fig 7/8 into the supplement.

-> Thank you very much for reviewing our manuscript and your important comments. We agree with your comments. We moved Fig7/8 to Supplementary fig and revised manuscript.

Minor

Typos and grammar remain major issues:

e.g. Sentence in lines 299-301: "Thus, in this study, we focused on whether...."

e.g. 310-312:" In vivo analysis consistently demnstarted whereas both...

-> I am sorry for the typing error. Although whole manuscript was checked by the English editing service company "Enago (Crimson Interactive Pvt. Ltd)", we checked again throughout the manuscript.

Reviewer #2 (Remarks to the Author):

The authors have addressed my comments and the revised manuscript is improved.

-> Thank you very much for reviewing our manuscript and your comments.